# CAT-LLM: Context-Aware Training Enhanced Large Language Models for Multi-modal Contextual Image Retrieval

## Abstract

Recently, the unprecedented advancement of Large Language Models (LLMs) has revolutionized in numerous applications in the vision-language domain. Inspired by the extraordinary visual understanding and logical reasoning abilities, we propose a method that employs LLMs to address the Multi-Modal Contextual Image Retrieval (MMCIR) problem, where the input hints include both visual and textual queries. Specifically, given a query comprising a sequence of images and texts, MMCIR aims to select an image from a gallery that best matches the context of the query. In this paper, we first construct a Multi-Modal Captioning (MMC) dataset by enriching existing image captioning datasets from ⟨image, caption⟩ to ⟨reference image, reference caption, text condition, target caption⟩. Then, we introduce a Context-Aware Captioning (CA-Cap) and a Context-Aware Text Matching (CA-TM) objective to instruct a frozen LLM for MMCIR. These specialized objectives enable the LLM to better understand multi-modal inputs and output visual representation from complex multi-modal contexts. Comprehensive experiments demonstrate the effectiveness of our method on recent Zero-Shot Composed Image Retrieval (ZS-CIR) benchmarks (i.e., CIRCO, CIRR, and GeneCIS), and in complex scenarios with dense multi-modal inputs like Visual Storytelling and Visual Dialog.

## 1 Introduction

In this paper, we investigate the problem of Multi-Modal Contextual Image Retrieval (MMCIR), in which the goal is to retrieve target image(s) using various contextual inputs, encompassing both images and text in a flexible manner. One particular and significant case is the Composed Image Retrieval (CIR) task, which aims to retrieve a target image given a multi-modal query consisting of a reference image and a text condition (i.e., relative caption). Early works (Baldrati et al., 2022a;b; Delmas et al., 2022; Lee et al., 2021; Liu et al., 2021b) train CIR models in supervised learning to combine the reference image and relative caption. Recent advancements (Saito et al., 2023; Baldrati et al., 2023; Vaze et al., 2023; Liu et al., 2023) have shifted the focus towards Zero-Shot Composed Image Retrieval (ZS-CIR) task, which aspires to perform CIR task without relying on human-annotated triplets. These methods either train a text-inversion network to combine reference image and text conditions into CLIP(Radford et al., 2021) text encoder, or automatically build triples from image-text pairs to train CIR models.

Despite the promising results shown in ZS-CIR benchmarks, these methods have inherent limitations: (1) **Relative-Caption Constraint**: These methods all utilize text encoders derived from image-text matching models (e.g., CLIP text encoder) to handle the text condition (i.e., relative caption). However, studies have shown that such text encoders struggle with understanding object relations, word order and logic (Yuksekgonul et al., 2022; Ma et al., 2023; Thrush et al., 2022; Wang et al., 2023), thus limiting their application in free-text condition scenarios. (2) **Single-Input Constraint**: Existing CIR methods are designed to handle queries consisting of only one image and one text condition, which restricts their applicability in a more complex multi-modal scenario that may involve multiple images and texts.

To address these challenges, we propose to incorporate a Large Language Model (LLM) into the MMCIR problem for several reasons: (1) LLMs excel at understanding not just caption-style conditions but also free-text conditions; (2) LLMs are skilled at processing and integrating contextual information; and (3) LLMs can manage inputs with extensive context. The subsequent challenge lies in integrating visual inputs into the LLM and extracting visual information from the LLM context. Koh et al. (2023a;b) enable a frozen language model to process and output images by leveraging (a) an image captioning task to learn to process visual input, and (b) an image-text matching task to learn to produce visual representation. By training on image-caption pairs sourced from the web, the LLM gains the ability to process multi-modal inputs and outputs. However, empirical evaluations reveal that the multi-modal LLM, built in this manner, doesn't achieve the desired performance on current ZS-CIR benchmarks. It suggests that the multi-modal ability learned only from image-text pairs is inadequate for scenarios demanding an in-depth understanding of multi-modal context.

In this work, we instruct a frozen LLM to MMCIR via context-aware training. To this end, we first introduce a Multi-Modal Captioning (MMC) dataset, which is automatically constructed by an off-the-shelf LLM. Given a web-collected ⟨reference image, reference caption⟩ pair, we first input the reference caption to an LLM and prompt the LLM to produce a free-text condition. In the subsequent step, the LLM is asked to generate a target image caption, taking into account both the initial reference caption and the generated text condition. Notably, as this data generation procedure is text-only, it leverages the LLM's inherent strengths in text generation and logical reasoning, resulting in a diverse and coherent dataset. We generate 1 million 4-tuples, i.e., ⟨reference image, reference caption, text condition, target caption⟩, using reference image-caption pairs from CC3M (Sharma et al., 2018).

Given the proposed MMC dataset, we introduce context-aware training to instruct a frozen LLM to MMCIR. We propose two tasks namely Context-Aware Captioning (CA-Cap) and Context-Aware Text Matching (CA-TM) to learn map visual inputs into LLM space and output visual representation from LLM multi-modal context. Specifically, given the input comprising a reference image and a text condition, a frozen LLM is tasked either with generating a target caption (i.e., CA-Cap) or with producing a visual representation to retrieve the target caption (i.e., CA-TM). Compared to the conventional image-captioning and image-text matching training, our proposed context-aware training, on the one hand, refines the visual mapping process by incorporating textual context, effectively integrating visual features into the language model's semantic space. On the other hand, the model is trained to extract visual representation from complex multi-modal context, rather than simply condensing text context into a visual representation (i.e., image-text matching training). Our main contributions are as follows:

(1) We propose CAT-LLM, a Context-Aware Training enhanced LLM for multi-modal contextual image retrieval. Trained with proposed context-aware captioning (CA-Cap) and context-aware text matching (CA-TM) objectives, CAT-LLM adapts well to various MMCIR scenarios.

(2) We construct a Multi-Modal Captioning (MMC) dataset containing 1 million tuples of 4 elements ⟨reference image, reference caption, text condition, target caption⟩. In this work, we utilize MMC for context-aware training, demonstrating its effectiveness in enhancing multi-modal understanding.

(3) We evaluate CAT-LLM on various MMCIR scenarios. We achieve competitive results on ZS-CIR benchmarks, i.e., CIRCO, CIRR and GeneCIS, demonstrating the potential of LLM in this field. In scenarios involving multiple images and texts inputs, i.e., Visual Storytelling and Visual Dialog, CAT-LLM consistently outperforms other LLM-based retrieval approaches, further demonstrating the effectiveness of our proposed context-aware training.

## 2 RELATED WORK

**LLMs for Vision-Language Tasks.** Recently, there have been many efforts that apply LLMs to vision-language tasks. Li et al. (2023); Alayrac et al. (2022); Zhu et al. (2023) integrate visual input into LLMs leveraging an adaptor or cross-attention mechanism. With the LLM's robust textual capabilities, they can perform not only conventional vision-language generation tasks like image captioning and visual question answering but also extend to more complex applications such as visual dialog and visual story generation. Another line of work (Koh et al., 2023a;b) further explore the potential of leveraging LLMs for the contextual image retrieval task. In this paper, we utilize

an LLM to: (1) generate a multi-modal captioning dataset for context-aware training; (2) handle the multi-modal input and extract a visual representation from LLM multi-modal context.

**Zero-Shot Composed Image Retrieval (ZS-CIR).** The goal of ZS-CIR is to perform the Composed Image Retrieval (CIR) task without requiring labeled triplets for training. Pic2Word (Saito et al., 2023) and CIRCO (Baldrati et al., 2023) map the input image to pseudo text tokens in order to flexibly compose image and text queries in CLIP text-encoder. Another stream of research (Vaze et al., 2023; Gu et al., 2023; Liu et al., 2023) automatically construct CIR triplets from widely available image-caption pairs for CIR model training. Specifically, Vaze et al. (2023) convert image captions into scene-graph representations and then use the relationships between scene-graphs to construct text conditions between images. Gu et al. (2023) construct 18M triplets by leveraging LLM and Stable Diffusion (Rombach et al., 2022). Liu et al. (2023) edit the reference image caption with pre-defined sentence templates or adopt large-language models to generate a target caption. In this work, we leverage LLM to perform zero-shot composed image retrieval. Briefly, we map the input image to LLM embedding space to better compose the input image and text condition. Furthermore, we construct ⟨reference image, text condition, target image caption⟩ triplets and employ a pseudo CIR training to improve both visual mapping and text-condition comprehension.

## 3 METHOD

This section delineates our multi-modal contextual image retrieval framework, as illustrated in Figure 1. Section 3.1 details the generation process of our Multi-Modal Captioning dataset. In Section 3.2, we introduce how to enable a frozen Large Language Model (LLM) to process visual input. In Section 3.3, we introduce how to extract visual representation from LLM multi-modal context.

### 3.1 DATA GENERATION

Web-collected weakly paired image-caption data (Sharma et al., 2018; Schuhmann et al., 2022) has demonstrated its effectiveness in image-text retrieval tasks (Radford et al., 2021), due to its vast scale and diversity. However, collecting paired data for multi-modal contextual retrieval, which demands a multi-modal contextual query and a target image, poses significant challenges. The absence of large-scale training data limits the development of multi-modal contextual image retrieval.

To address this, we propose to automatically generate a large-scale multi-modal captioning dataset from existing ⟨image, caption⟩ pairs by leveraging an off-the-shelf LLM. Given a reference image $I_{\text{ref}}$ and its associated caption $T_{\text{refc}}$, we input the $T_{\text{refc}}$ into an LLM along with a task-specific prompt. The LLM then generates a free-text condition $T_{\text{con}}$, which could serve as an editing order to alter attributes and objects, or describe the differences between the reference image and the target image, etc. Following this, the LLM is tasked with generating a target caption $T_{\text{tgtc}}$, incorporating both the reference caption $T_{\text{refc}}$ and the newly generated text condition $T_{\text{con}}$. As a result, we derive a $\langle I_{\text{ref}}, T_{\text{refc}}, T_{\text{con}}, T_{\text{tgtc}} \rangle$ tuple from a $\langle I_{\text{ref}}, T_{\text{refc}} \rangle$ pair, which can be automatically collected from the web.

To enhance the performance of the LLM (i.e., Llama2 (Touvron et al., 2023)) in data generation, we utilize the in-context learning techniques. Specifically, we employ a state-of-the-art LLM, GPT-4[1], to generate 20 in-context examples. In practice, we find that GPT-4 can effectively understand our data generation task and produce diverse samples. During each sample generation process, we provide Llama2 with a task description and randomly select one in-context example as the task-specific prompt. This approach ensures diverse and high-quality generated samples. More details are provided in Appendix C.

### 3.2 MAPPING VISUAL INPUT TO LLM

**Visual Mapping.** Following the latest advancements in vision-LLM research (Mokady et al., 2021; Merullo et al., 2022; Koh et al., 2023b), we employ a linear mapping layer (i.e., an adaptor) that maps CLIP visual features into the LLM's embedding space. Given an input image $\mathbf{I}$, we first utilize a frozen CLIP image encoder, denoted as $E_{\text{image}}$, to extract its visual feature. Subsequently, a

---

[1]`https://openai.com/research/gpt-4`

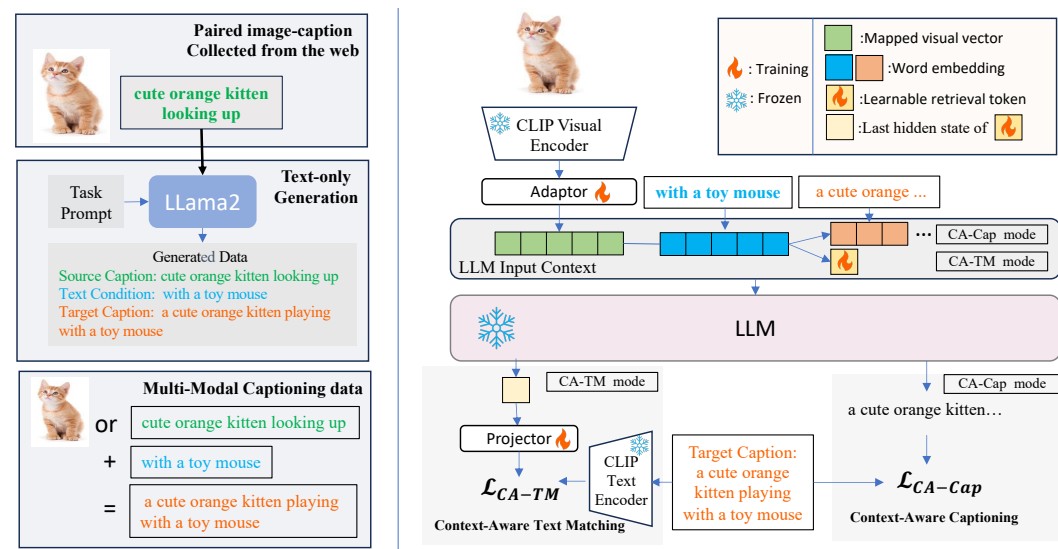

Figure 1: Framework of CAT-LLM. Given ⟨image, caption⟩ pairs, we feed the caption along with a task prompt to Llama2 to generate both a text condition and a target caption. By integrating the initial image with the generated text condition and target caption, we facilitate context-aware training, enabling a frozen language model to handle multi-modal inputs and output visual representations.

linear mapping layer, represented as $f_{\text{map}}$, is applied to map this visual feature into the LLM textual embedding space, yielding n visual vectors $\mathbf{V} = [\boldsymbol{v_0}, \boldsymbol{v_1}, ..., \boldsymbol{v_n}] = f_{\text{map}}(E_{\text{image}}(\mathbf{I}))$. The visual vector's dimension matches the LLM's word embedding dimension.

**Naive Image Captioning Objective.** Previous methods leverage a conventional image captioning objective to train this mapping layer by predicting the next token conditioned on both the visual tokens and previous caption tokens. The objective can be formulated as:

$$\mathcal{L}_{\text{Cap}}(\theta_m) = -\frac{1}{|t|} \sum_{i=1}^{|t|} \log P\Big(t_i | f_{\text{map}}(E_{\text{image}}(\mathbf{I})), \mathbf{t}_{<\mathbf{i}}\Big), \tag{1}$$

where $t_i$ represents the $i_{th}$ token of the caption, $\theta_m$ denotes the weight of mapping layer $f_{\text{map}}$ and $P$ denotes the frozen language model.

**Context-Aware Captioning (CA-Cap) Objective.** In this work, we enhance this mapping by incorporating it with the generated dataset described in Section 3.1. Given triplet $\langle I_{\text{ref}}, T_{\text{con}}, T_{\text{tgtc}} \rangle$, the LLM learns to predict the next token conditioned on the visual vectors $\mathbf{V}$, text conditions tokens and previous target caption tokens. The context-aware captioning objective is described as:

$$\mathcal{L}_{\text{CA-Cap}}(\theta_m) = -\frac{1}{|t|} \sum_{i=1}^{|t|} \log P\Big(t_i | f_{\text{map}}(E_{\text{Image}}(\mathbf{I}_{\text{ref}})), \mathbf{c_1}, \mathbf{c_2}, .., \mathbf{c_{|c|}}, \mathbf{t}_{<\mathbf{i}}\Big), \tag{2}$$

where $c_i$ denotes the $i_{th}$ token of text condition and $t_i$ denotes the $i_{th}$ token of target caption.

Compared to the conventional image captioning objective (i.e., Equation 1), our proposed objective offers distinct advantages. **Enhanced Linguistic Visual Mapping**: Our training objective refines the mapping process by incorporating textual cues (i.e., the text condition), leading to a textual-aware mapping that effectively integrates visual features into the language model's semantic space. **Enhanced Textual Interaction**: During the training process, the language model is required to query the mapped visual vectors based on the text condition to derive the target caption. This ensures the mapped visual vectors are optimized to support textual queries.

### 3.3 EXTRACTING VISUAL OUTPUT FROM LLM

In this section, we introduce how to extract visual representation from LLM. Following (Koh et al., 2023b), we leverage a learnable token to extract visual information from the LLM multi-modal

context. Specifically, given LLM contexts $\mathbf{Z} = \mathbf{z_1}, \mathbf{z_2}, ..., \mathbf{z_i}$, where $\mathbf{z_i}$ can be both a mapped visual vector or natural language token, we add a $ret$ token after input context to capture visual information from the multi-modal context. The last hidden state of the $ret$ is used to output visual representation.

**Naive Image-Text Matching (ITM) Objective.** FROMAGe (Koh et al., 2023b) use an image-text matching objective to train the $ret$ embedding. Specifically, given a paired image and caption, they append the $ret$ token after the caption tokens as input to the LLM. The last hidden state of $ret$ is used as the LLM's output, denoted as $h(ret|C)$, where $C$ denotes the caption tokens. $h(ret|C)$ is then projected to CLIP's latent space through a simple linear layer, represented as $p = f_{\text{proj}}(h(ret|C))$. An infoNCE (Oord et al., 2018) loss is employed to align the projected embedding and its CLIP visual feature. The objective can be formulated as:

$$\mathcal{L}_{\text{ITM}}(ret, \theta_p) = -\frac{1}{N} \sum_{i=1}^{N} \left( \log \frac{\exp(sim(f_{\text{proj}}(h(ret|C_i)), \mathbf{e_i})/\tau)}{\sum_{j=1}^{N} \exp(sim(f_{\text{proj}}(h(ret|C_i)), \mathbf{e_j})/\tau)} \right), \quad (3)$$

where $\theta_p$ denotes the weight of project layer $f_{\text{proj}}$, $\mathbf{e_i} = \mathbf{E}_{\text{image}}(\mathbf{I_i})$ and $sim$ denotes cosine similarity function.

**Context-Aware Text Matching (CA-TM) Objective.** Trained with naive image-text matching objective, $ret$ bridges the LLM context with the CLIP feature space. However, in this case, the $ret$ token primarily functions as a text summarization token, condensing the text context into the CLIP feature space, lacking the ability to selectively extract target information based on the given multi-modal context. To this end, we introduce our context-aware text matching objective to enhance the multi-modal contextual retrieval ability. Given triplet $\langle I_{\text{ref}}, T_{\text{con}}, T_{\text{tgtc}} \rangle$, we input the $I_{\text{ref}}$ and $T_{\text{con}}$ to the LLM with $ret$ attached at the end. In this scenario, $ret$ learns to extract target information from the multi-modal context to match the target caption $T_{\text{tgtc}}$. The objective can be formulated as:

$$\mathcal{L}_{\text{CA-TM}}(ret, \theta_p, \theta_m) = -\frac{1}{N} \sum_{i=1}^{N} \left( \log \frac{\exp(sim(f_{\text{proj}}(h(ret|\mathbf{V_i}, C_i)), \mathbf{e_i})/\tau)}{\sum_{j=1}^{N} \exp(sim(f_{\text{proj}}(h(ret|\mathbf{V_i}, C_i)), \mathbf{e_j})/\tau)} \right), \quad (4)$$

where $\mathbf{V}$ denotes the mapped visual vectors of $I_{\text{ref}}$, $C$ denotes the language tokens of $T_{\text{con}}$ and $\mathbf{e}$ denotes the CLIP text feature of $T_{\text{tgtc}}$ here. It should be noted that the context-aware text matching objective also optimizes the visual mapping, as it is related to the visual input $\mathbf{V}$.

Training with a Context-Aware Text Matching objective brings the following benefits: (a) The $ret$ token is trained under multi-modal contexts, making it better adapted to handle multi-modal inputs effectively. (b) The $ret$ token learns to selectively extract information based on the multi-modal context, rather than indiscriminately condensing all the input.

## 3.4 TRAINING AND INFERENCE

Given the tuples of 4 elements $\langle$reference image, reference caption, text condition, target caption$\rangle$ from MMC, we combine the four aforementioned losses with corresponding weight $\lambda$ to train the CAT-LLM. The combined loss function is expressed as:

$$\mathcal{L}_{\text{overall}} = \lambda_{\text{Cap}}\mathcal{L}_{\text{Cap}} + \lambda_{\text{ITM}}\mathcal{L}_{\text{ITM}} + \lambda_{\text{CA-Cap}}\mathcal{L}_{\text{CA-Cap}} + \lambda_{\text{CA-TM}}\mathcal{L}_{\text{CA-TM}}, \quad (5)$$

The first two losses are based on the $\langle$reference image, reference caption$\rangle$ pairs, while the latter two context-aware losses are based on the triplets $\langle$reference image, text condition, target caption$\rangle$.

During inference, we feed the input image and text into the LLM, appending the $ret$ token afterwards to extract a visual representation from the multi-modal input, denoted as **CAT-LLM-(ret)**. As an alternative, we can leverage the LLM to autoregressively generate a caption for the multi-modal input. The caption is then used to obtain a representation from the CLIP text-encoder, denoted as **CAT-LLM-(cap)**. Finally, we introduce a simple fusion method that sums the last output representation of these two approaches, denoted as **CAT-LLM-(ret+cap)**. For a clearer understanding, we provide a detailed illustration of the CAT-LLM inference process in Appendix E.

**Implementation Details.** Following Koh et al. (2023b), we use the OPT-6.7B (Zhang et al., 2022) model as our LLM backbone. We employ CLIP ViT-B/16 or ViT-L/14 as our image-text matching model. The CLIP visual feature is mapped to 4 visual-language tokens through a single linear layer. The model is trained on MMC for 20000 iterations with a batchsize of 120. Both the LLM and CLIP model are frozen. The loss weights in Equation 5 are set to 1.

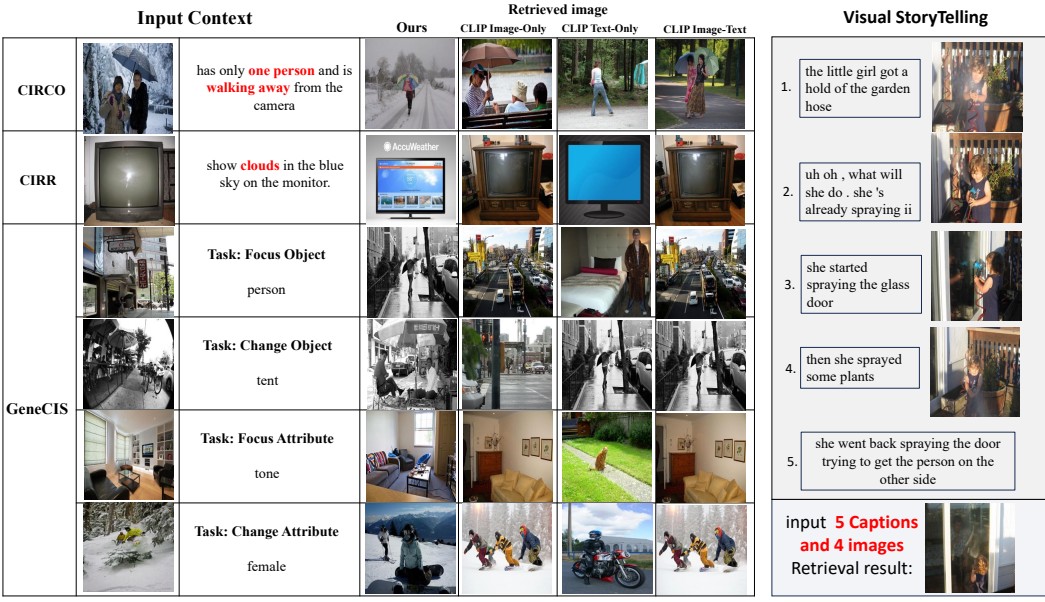

Figure 2: Example of image retrieval task with varying multi-modal context inputs. CAT-LLM can handle scenarios with a single image and single text condition inputs, as well as adapt to tasks like visual storytelling which involve multiple continuous image-text inputs. By designing specific prompts, CAT-LLM can address unique task settings, such as those in GeneCIS. The prompts designed for different tasks are shown in Appendix D.

## 4 EXPERIMENTS

As shown in Figure 2, when trained on our multi-modal captioning dataset using proposed context-aware objectives, CAT-LLM can effectively retrieve the target image with arbitrary multi-modal input. In this section, we conduct extensive experiments on two multi-modal contextual image retrieval tasks, namely zero-shot composed image retrieval (ZS-CIR) (in Section 4.1) and dense multi-modal contextual retrieval (in Section 4.2). Ablation studies on our proposed context-aware objectives are in Section 4.3.

### 4.1 ZERO-SHOT COMPOSED IMAGE RETRIEVAL (ZS-CIR)

**Benchmarks and Metrics.** We evaluate CAT-LLM on three ZS-CIR benchmarks: CIRCO (Baldrati et al., 2023), CIRR (Liu et al., 2021a) and GeneCIS (Vaze et al., 2023). CIRCO is a new open-domain ZS-CIR benchmark with multiple annotated ground truths. For CIRCO evaluation, we report the fine-grained metric of mean Average Precision (mAP@$K$). The mAP@$K$ metrics are computed considering all the ground truth images for each query. For CIRR and GeneCIS evaluations, we report the Recall@$K$ metric.

**Baselines and Competing Methods.** We compare our approach with several zero-shot baselines and recent ZS-CIR methods, including: (1) *Image-only*: The CLIP visual feature of the reference image is used to retrieve the target image. (2) *Text-only*: The CLIP text feature of the text condition is used to retrieve the target image. (3) *Image+Text*: The CLIP visual feature of the reference image and the CLIP text feature of the text condition are summed together to retrieve the target image. (4) CLIP-based textual inversion methods: *Pic2Word* (Saito et al., 2023) and *SEARLE* (Baldrati et al., 2023). (5) Combiner (Baldrati et al., 2022a) trained on constructed triplets: **Conbiner-(GeneCIS)**(Vaze et al., 2023), *CompoDiff* (Gu et al., 2023) and *TransAgg* (Liu et al., 2023). (6) LLM-based retrieval method: *FROMAGe* (Koh et al., 2023b) and **CAT-LLM** (ours).

**Analysis on CIRCO.** Table 1 shows the result on CIRCO test set. CAT-LLM-(ret+cap) achieves the best performance on all metrics, outperforming prior CLIP-based zero-shot methods and LLM-based methods. FROMAGe achieves inferior performance, indicating that learning from ⟨image, caption⟩ pairs is insufficient to handle challenging multi-modal scenarios. Benefiting from our context-aware

Table 1: Quantitative results on CIRCO test.

| Backbone | Method | mAP@K | | | |
|---|---|---|---|---|---|
| | | $K=5$ | $K=10$ | $K=25$ | $K=50$ |
| L/14 | Image-only | 2.79 | 3.18 | 3.75 | 4.12 |
| | Text-only | 2.50 | 2.64 | 3.11 | 3.38 |
| | Image + Text | 6.37 | 7.04 | 8.11 | 8.72 |
| | Pic2Word | 8.72 | 9.51 | 10.64 | 11.29 |
| | SEARLE | 11.68 | 12.73 | 14.33 | 15.12 |
| | FROMAGe | 4.0 | 4.44 | 5.26 | 5.73 |
| | CAT-LLM-(cap) | 6.43 | 6.84 | 7.77 | 8.30 |
| | CAT-LLM-(ret) | 13.55 | 14.70 | 16.35 | 17.28 |
| | CAT-LLM-(ret+cap) | **15.00** | **15.73** | **17.51** | **18.45** |
| B/16 | Image-only | 1.30 | 1.74 | 2.21 | 2.52 |
| | Text-only | 2.59 | 2.75 | 3.12 | 3.30 |
| | Image + Text | 2.60 | 3.19 | 4.12 | 4.63 |
| | CAT-LLM-(cap) | 5.35 | 5.59 | 6.32 | 6.68 |
| | CAT-LLM-(ret) | 12.79 | 13.28 | 14.89 | 15.65 |
| | CAT-LLM-(ret+cap) | **13.95** | **14.47** | **16.00** | **16.74** |

Table 2: Quantitative results on CIRR test set.

| Backbone | Method | Recall@K | | | | Recall$_{Subset}$@K | | |
|---|---|---|---|---|---|---|---|---|
| | | $K=1$ | $K=5$ | $K=10$ | $K=50$ | $K=1$ | $K=2$ | $K=3$ |
| L/14 | Image-only | 7.13 | 23.04 | 32.99 | 56.63 | 20.55 | 40.96 | 61.04 |
| | Text-only | 20.55 | 44.17 | 55.95 | 78.94 | 60.74 | 80.38 | 90.72 |
| | Image+Text | 12.27 | 35.81 | 48.48 | 77.04 | 33.33 | 57.78 | 75.95 |
| | TransAgg | 25.04 | 53.98 | 67.59 | 88.94 | 55.33 | 76.82 | 88.94 |
| | CompoDiff | 18.24 | 53.14 | **70.82** | 90.35 | - | - | - |
| | Pic2Word | 23.90 | 51.70 | 65.30 | 87.80 | - | - | - |
| | SEARLE | 24.22 | 52.41 | 66.29 | 88.63 | 53.71 | 74.63 | 87.61 |
| | FROMAGe | 10.96 | 31.40 | 44.33 | 72.97 | 34.07 | 58.84 | 76.80 |
| | CAT-LLM-(cap) | 20.68 | 45.13 | 56.96 | 79.61 | 62.24 | 81.25 | 90.75 |
| | CAT-LLM-(ret) | 22.65 | 53.16 | 66.43 | 90.07 | 57.33 | 77.88 | 89.93 |
| | CAT-LLM-(ret+cap) | **27.21** | **57.27** | 70.24 | **90.70** | **63.18** | **82.39** | **92.12** |
| B/16 | Image-only | 6.56 | 21.33 | 29.71 | 52.51 | 20.72 | 41.01 | 60.87 |
| | Text-only | 20.87 | 46.15 | 58.29 | 80.63 | 61.98 | 81.18 | 91.01 |
| | Image+Text | 12.46 | 35.90 | 48.77 | 77.35 | 32.96 | 56.65 | 75.11 |
| | CAT-LLM-(cap) | 20.48 | 44.51 | 56.87 | 80.36 | 62.48 | 81.08 | 90.70 |
| | CAT-LLM-(ret) | 23.18 | 53.64 | 67.16 | 90.41 | 57.86 | 79.13 | 90.00 |
| | CAT-LLM-(ret+cap) | **27.88** | **57.86** | **70.92** | **91.59** | **64.31** | **82.94** | **91.49** |

training, CAT-LLM-(ret) understands the text condition, and efficiently extracts information from the multi-modal context to retrieve the target image. CAT-LLM-(cap) underperforms on CIRCO, suggesting that only using generative ability of the language model to extract information from the multi-modal context leads to suboptimal results. By simply summing the output of the two approaches, the performance further improves, indicating that CAT-LLM-(ret) and CAT-LLM-(cap) have different preferences when extracting information from the multi-modal context. We provide more qualitative results and analysis on CIRCO validation set in Appendix A.4.

**Analysis on CIRR.** Table 2 shows the results on CIRR test set. Notably, the CIRR benchmark has a strong bias towards the text modality input (Saito et al., 2023; Baldrati et al., 2023). The Text-only baseline surpasses the Image+Text baseline a lot and even outperforms most ZS-CIR methods on the $Recall_{subset}@K$ metrics. CAT-LLM-(ret+cap) outperforms other methods on most metrics. CAT-LLM-(cap) performs better on CIRR compared to CIRCO. This suggests that CAT-LLM-(cap) has a better capability to extract textual information, due to its inherent use of a language model for caption generation, showing a preference for text modality input.

**Analysis on GeneCIS.** GeneCIS introduces four unique tasks: Focus Attribute, Change Attribute, Focus Object, and Change Object. For each task, only a single object name or attribute name is provided. This setup differs significantly from prior benchmarks such as CIRR and CIRCO, which often provide caption-style text conditions. Table 3 shows the results. CAT-LLM-(ret) is competitive with Combiner (GeneCIS), which is trained on triplets crafted from the four tasks. Notably, in 'Change Attribute' and 'Focus Attribute' settings, only the attribute is provided without further specifics. For instance, when given the attribute 'color', it's ambiguous whether it refers to the background or a specific entity's color. The ambiguous text condition confused our model, leading to less impressive results.

Table 3: Quantitative results on GeneCIS. $^\dagger$ indicates that CLIP model is **not frozen**.

| | | Focus Attribute | | | Change Attribute | | | Focus Object | | | Change Object | | | |
|---|---|---|---|---|---|---|---|---|---|---|---|---|---|---|
| | | R@1 | R@2 | R@3 | R@1 | R@2 | R@3 | R@1 | R@2 | R@3 | R@1 | R@2 | R@3 | Avg R@1 |
| B/16 | Image Only | 18.1 | 30.1 | 40.6 | 11.5 | 21.9 | 30.9 | 9.4 | 17.0 | 25.4 | 7.6 | 17.1 | 25.5 | 11.7 |
| | Text Only | 10.3 | 20.9 | 30.4 | 10.2 | 18.2 | 26.1 | 7.4 | 14.0 | 23.0 | 8.1 | 16.4 | 24.7 | 9.0 |
| | Image + Text | 17.1 | 29.5 | 40.5 | 13.1 | 22.2 | 31.9 | 11.5 | 20.1 | 29.2 | 9.8 | 20.0 | 28.9 | 12.9 |
| | Combiner (GeneCIS)$^\dagger$ | **19.7** | **31.7** | **42.1** | **16.2** | **27.3** | **37.5** | 16.6 | 27.7 | 37.2 | 18.0 | 32.2 | 41.6 | 17.6 |
| | CAT-LLM-(cap) | 14.1 | 25.3 | 35.3 | 10.6 | 21.4 | 30.1 | 11.5 | 20.8 | 28.8 | 11.0 | 21.1 | 29.6 | 11.8 |
| | CAT-LLM-(ret) | 19.0 | 30.4 | 40.2 | 14.9 | 25.2 | 32.5 | **17.6** | **28.4** | 37.4 | **19.2** | **32.7** | **42.5** | **17.7** |
| | CAT-LLM-(ret+cap) | 18.4 | 30.5 | 40.3 | 15.7 | 25.6 | 33.8 | 16.1 | **28.4** | **37.6** | 18.0 | 30.8 | 41.7 | 17.1 |
| L/14 | Image Only | 18.2 | 29.6 | 40.0 | 9.2 | 20.2 | 29.1 | 9.6 | 16.2 | 25.5 | 6.8 | 16.0 | 24.7 | 11.0 |
| | Text Only | 12.3 | 20.2 | 31.3 | 8.1 | 17.7 | 24.6 | 8.2 | 15.3 | 24.1 | 7.6 | 15.4 | 25.1 | 9.1 |
| | Image+Text | 17.6 | 29.5 | 40.0 | 10.6 | 22.1 | 31.9 | 11.8 | 21.4 | 29.0 | 10.3 | 21.0 | 31.1 | 12.6 |
| | FROMAGe | **19.2** | **31.1** | 40.5 | 12.2 | 21.7 | 30.5 | 13.0 | 24.5 | 33.2 | 12.9 | 24.7 | 32.9 | 14.3 |
| | CAT-LLM-(cap) | 12.0 | 24.1 | 34.1 | 13.3 | 23.1 | 31.7 | 14.7 | 24.2 | 33.7 | 10.9 | 20.5 | 29.0 | 12.7 |
| | CAT-LLM-(ret) | 18.5 | 30.2 | 40.7 | **15.2** | 25.6 | 34.3 | **15.3** | 25.4 | 34.6 | **17.2** | **27.0** | **37.5** | **16.6** |
| | CAT-LLM-(ret+cap) | 17.6 | 29.3 | **41.0** | 15.1 | **25.7** | 34.5 | 14.6 | **27.2** | **36.3** | 15.9 | 26.8 | 36.7 | 15.8 |

**Conclusions on ZS-CIR benchmarks.** (1) CAT-LLM achieves remarkable performance across existing ZS-CIR benchmarks, outperforming prior baselines and methods that do not incorporate LLM. It showcases the potential of applying LLM in the CIR task. (2) CAT-LLM notably adapts to a variety of text conditions. In scenarios like the GeneCIS evaluation, which presents unique text conditions that differ from traditional relative captions, we can design task-specific prompts to effectively adapt to different tasks. This highlights CAT-LLM's adaptability to a wide range of text conditions. (3) CAT-LLM-(ret) and CAT-LLM-(cap) show different preferences to multi-modal

context information. CAT-LLM-(ret) excels in selectively extracting information from multi-modal contexts, while CAT-LLM-(cap) tends to text modality information. CAT-LLM can adapt to different scenarios by dynamically choosing the output mode based on the given input. At the same time, it indicates that our retrieval token struggles to **adaptively** extract information from the multi-modal context, which is the direction of our future improvements.

## 4.2 DENSE MULTI-MODAL CONTEXTUAL IMAGE RETRIEVAL

To investigate CAT-LLM's multi-modal contextual retrieval ability in more complex scenarios, we consider the dense multi-modal contextual retrieval task where the input encompasses multiple images and texts. Following Koh et al. (2023b), we evaluate the zero-shot dense multi-modal image retrieval ability on Visual Storytelling (Huang et al., 2016) and Visual Dialogue (Das et al., 2017). In this section, we only report the results of CAT-LLM-(ret) for comparison with FROMAGe(Koh et al., 2023b) and GILL(Koh et al., 2023a).

**Visual Storytelling Results.** Each example in the Visual Storytelling (VIST) dataset comprises five temporally ordered image-text pairs, we report Recall@$K$ of the last image as metric. Following (Koh et al., 2023b), we explore several experimental settings featuring different input configurations: (1) single last caption as input; (2) input consisting of all five captions; (3) input incorporating five captions along with four associated images. Table 4 shows the results. CAT-LLM surpasses the image-text matching model CLIP as well as the LLM-based model FROMAGe and GILL(Koh et al., 2023a) in three richer contextual settings. Furthermore, by comparing the results of '1 caption' and '5 captions' settings, we observe that the performance of CLIP or FROMAGe doesn't see a significant improvement and may even decline from 5 captions. In contrast, our method largely benefits from this richer textual context. This underscores our method's robust capability in extracting information from extended contexts effectively.

Table 4: Zero-shot contextual image retrieval on Visual Storytelling. [†] indicates input images from the current story sequence are masked in the retrieval gallery.

| Model | Inputs | R@1 | R@5 | R@10 |
|---|---|---|---|---|
| CLIP ViT-L/14 | | 11.9 | **25.5** | **32.2** |
| FROMAGe | 1 caption | 11.3 | 24.6 | 32.1 |
| CAT-LLM | | 9.8 | 22.6 | 30.4 |
| CLIP ViT-L/14 | | 5.9 | 19.5 | 28.0 |
| FROMAGe | 5 captions | 10.8 | 23.8 | 31.7 |
| CAT-LLM | | **11.8** | **29.0** | **39.1** |
| CLIP ViT-L/14 | | 8.8 | 22.3 | 29.8 |
| FROMAGe | 5 captions[†] | 13.2 | 28.5 | 36.7 |
| CAT-LLM | | **13.8** | **31.2** | **40.5** |
| CLIP ViT-L/14 | | 2.4 | 21.3 | 34.0 |
| FROMAGe | | 18.2 | 42.7 | 51.8 |
| GILL | 5 captions, 4 images[†] | 20.3 | 45.0 | 53.7 |
| CAT-LLM | | **22.4** | **45.4** | **55.0** |

**Visual Dialog Results.** Each sample in Visual Dialog contains one image and a conversation about this image. We take the conversation as the input context to retrieve the corresponding image. Table 5 shows the text-to-image retrieval results on Visual Dialog. CAT-LLM consistently outperforms CLIP baseline and prior LLM-based retrieval methods. This demonstrates CAT-LLM's ability to extract visual representations from extensive text contexts, underscoring the effectiveness of our proposed context-aware training.

Table 5: Zero-shot text-to-image retrieval on Visual Dialog

| Model | R@1 | R@5 | R@10 |
|---|---|---|---|
| CLIP ViT-L/14 | 17.7 | 38.9 | 50.2 |
| FROMAGe | 20.8 | 44.9 | 56.0 |
| CAT-LLM | **24.8** | **50.1** | **63.2** |

## 4.3 ABLATIONS

**Ablations of proposed objectives.** Table 6 shows the result of the ablation study on training objectives. We can draw main conclusions that: (1) All of the four objectives have a positive effect on CAT-LLM; (2) Context-aware text matching (CA-TM) objective significantly improves the performance indicating that it helps the retrieval token learn to extract information from the multi-modal context. (4) A comparison between the 'w/o CA' and 'w/o CA-TM' results reveals that the Context-Aware Captioning (CA-Cap) objective markedly improves performance. It indicates that CA-Cap objective helps to learn a better mapping from visual space to LLM's space. Note that the CA-Cap objective is only used to optimize the visual adaptor.

**Visualizing the effectiveness of context-aware training** In Figure 3, we visualize the impact of proposed context-aware training to the LLM using a transformer explainable tool described in Chefer et al. (2021). As the figure shown, our context-aware training enables the model effectively

Table 6: Ablations of proposed objectives. Results of CAT-LLM-(ret) on CIRCO evaluation set.

| Model | $\mathcal{L}_{Cap}$ | $\mathcal{L}_{ITM}$ | $\mathcal{L}_{CA\text{-}Cap}$ | $\mathcal{L}_{CA\text{-}TM}$ | mAP@5 | mAP@10 | mAP@25 | mAP@50 |
|---|---|---|---|---|---|---|---|---|
| Base model | ✓ | ✓ | ✓ | ✓ | 13.84 | 14.09 | 14.87 | 15.20 |
| Only CA | | | ✓ | ✓ | 11.21 | 11.36 | 12.12 | 12.66 |
| w/o CA | ✓ | ✓ | | | 5.31 | 5.91 | 6.46 | 6.68 |
| w/o CA-TM | ✓ | ✓ | ✓ | | 8.13 | 8.41 | 8.87 | 9.12 |
| w/o CA-Cap | ✓ | ✓ | | ✓ | 13.53 | 13.73 | 14.45 | 14.79 |
| w/o ITM | ✓ | | ✓ | ✓ | 12.05 | 12.10 | 12.94 | 13.33 |
| w/o Cap | | ✓ | ✓ | ✓ | 12.58 | 13.22 | 14.15 | 14.29 |

composing the visual input and key textual cues to accurately retrieve the target image. For instance, in the first example, the model retrieves the target by identifying cues like 'same color', 'congested street', and 'stopped'. Conversely, the model without context-aware training tends to concentrate on visual tokens or with only partial textual cues, leading to incorrect retrieval results.

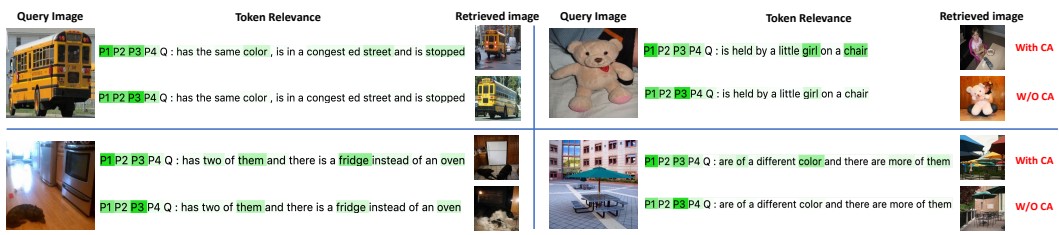

Figure 3: Visualization of the token relevance between retrieval token and the multi-modal context tokens. The deeper the green, the more significant the influence on the retrieval results. P1, P2, P3 and P4 denote the mapped visual tokens. CA denotes context-aware training.

**Can CAT-LLM benefit from stronger LLMs?** Table 7 shows the results of CAT-LLM with various LLM backbones on CIRCO test set. CAT-LLM effectively integrates different language models, gaining from increased model size and enhanced language capabilities. The powerful language model, LLama2, markedly enhances the performance of CAT-LLM. We provide more qualitative results on Appendix A.2 to illustrate the impact of a stronger language model backbone.

Table 7: CAT-LLM with various LLMs.

| LM | Mode | mAP@5 | mAP@10 | mAP@25 | mAP@50 |
|---|---|---|---|---|---|
| Opt-6.7B | (cap) | 6.43 | 6.84 | 7.77 | 8.30 |
| | (ret) | 13.55 | 14.70 | 16.35 | 17.28 |
| | (ret+cap) | 15.00 | 15.73 | 17.51 | 18.45 |
| Opt-2.7B | (cap) | 6.64 | 7.16 | 7.95 | 8.44 |
| | (ret) | 12.27 | 13.09 | 14.66 | 15.50 |
| | (ret+cap) | 14.05 | 14.87 | 16.52 | 17.35 |
| LLama2-7B | (cap) | 8.87 | 9.63 | 10.90 | 11.52 |
| | (ret) | 16.00 | 16.82 | 18.61 | 19.59 |
| | (cap+ret) | **17.28** | **18.27** | **20.22** | **21.17** |

**Can MMC dataset apply to conventional CIR methods?** Table 8 shows the results on CIRCO test set. We use MMC dataset to train a Combiner Baldrati et al. (2022a) model, which is a classic CIR method that employs a simple combiner component to integrate features from

Table 8: Results of Combiner trained on MMC

| Method | Training data | mAP@5 | mAP@10 | mAP@25 | mAP@50 |
|---|---|---|---|---|---|
| Pic2Word | CC3M | 8.72 | 9.51 | 10.64 | 11.29 |
| SEARLE | ImageNet1K | 11.68 | 12.73 | 14.33 | 15.12 |
| Combiner | MMC | 11.63 | 12.46 | 13.79 | 14.60 |
| CAT-LLM(ret) | MMC | 13.55 | 14.70 | 16.35 | 17.28 |

CLIP image encoder and text encoder. The Combiner model trained on the MMC dataset achieves comparable results, approaching previous textual inversion-based method SEARlE, demonstrating the effective of the generated MMC dataset. Although trained on the same MMC dataset, there is still a significant gap between the Combiner and CAT-LLM, demonstrating the effectiveness of LLM in MMCIR tasks. In Appendix A.1, we provide additional qualitative results and analyses on the significant differences between Combiner and CAT-LLM in processing logical words. The training details of Combiner are described in Appendix E.

## 5 CONCLUSION

In this paper, we investigated the capabilities of Large Language Models (LLM) for image retrieval tasks with multi-modal contextual queries. On recent ZS-CIR benchmarks, CAT-LLM surpasses approaches not utilizing an LLM, underscoring the promising potential of LLMs in this domain. Compared to previous LLM-based retrieval methods, our proposed context-aware training enhances the model's ability to handle multi-modal contexts. We hope that our context-aware training and multi-modal data generation strategies will inspire further exploration of LLMs for other vision-language tasks. In the future, we aim to delve deeper into enhancing the visual output mechanisms and also explore the implications of utilizing richer and more extensive training data.

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

## A    MORE QUALITATIVE RESULTS AND ANALYSIS

### A.1    ANALYSIS THE LOGICAL WORDS IN MMCIR TASKS

Different from conventional image-text retrieval, logical words like 'no' and 'instead of' occur more frequently in the multi-modal contextual image retrieval scenarios. These logical words pose a challenge to CIR models that rely on a CLIP text encoder to process textual queries. It is because the CLIP text encoder struggles in handling words like 'no' (Wang et al., 2023), which are critical to retrieve the correct target images. As shown in Figure 4, the CLIP based model Combiner struggles to comprehend text inputs containing logical words, leading incorrect retrieval results. In CAT-LLM, we leverage the aligned CLIP image-text space as the retrieval space. The CLIP text encoder is utilized to process the target caption in MMC and the generated caption from CAT-LLM-(cap). In these two scenarios, the text inputs of CLIP text encoder are more likely in a caption-style. The text conditions, which are more likely to contain logical words, are processed in the language model space, where the logical words like 'not' and 'instead of' can be easily understood.

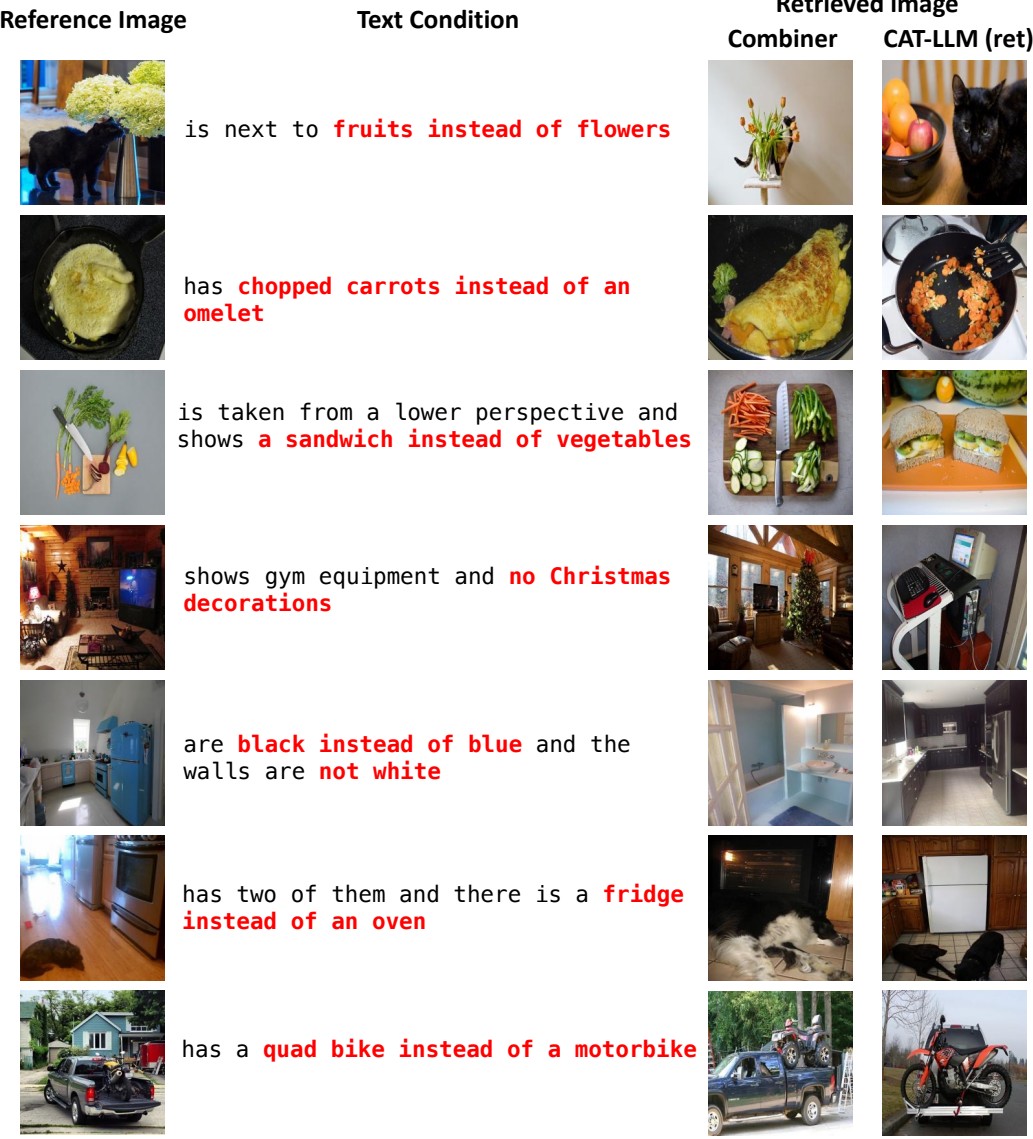

Figure 4: Examples from CIRCO validation set containing logical words. Both Combiner and CAT-LLM are trained on proposed MMC dataset.

## A.2 QUALITATIVE RESULTS OF CAT-LLM WITH VARIOUS LM BACKBONES

In Table 7, we quantitatively demonstrate that CAT-LLM benefits from a more advanced Language Model Llama2. In this section, we provide additional qualitative results to analyze the impact of a stronger language model on CAT-LLM. We select some hard samples from CIRCO test set and list the results of CAT-LLM with Opt-6.7B and Llama2-7B. As shown in Figure 5, every test sample has a complex text condition, requiring comprehensive multi-modal abilities to retrieve the correct image. Overall, benefiting from a more powerful language model, CAT-LLM with Llama2-7B shows improved performance in understanding these complex multi-modal inputs. For instance, in the first example, CAT-LLM (Llama2-7B) identifies critical textual cues such as 'white', 'without a wind shield', and 'a similar shape', effectively retrieving the correct image.

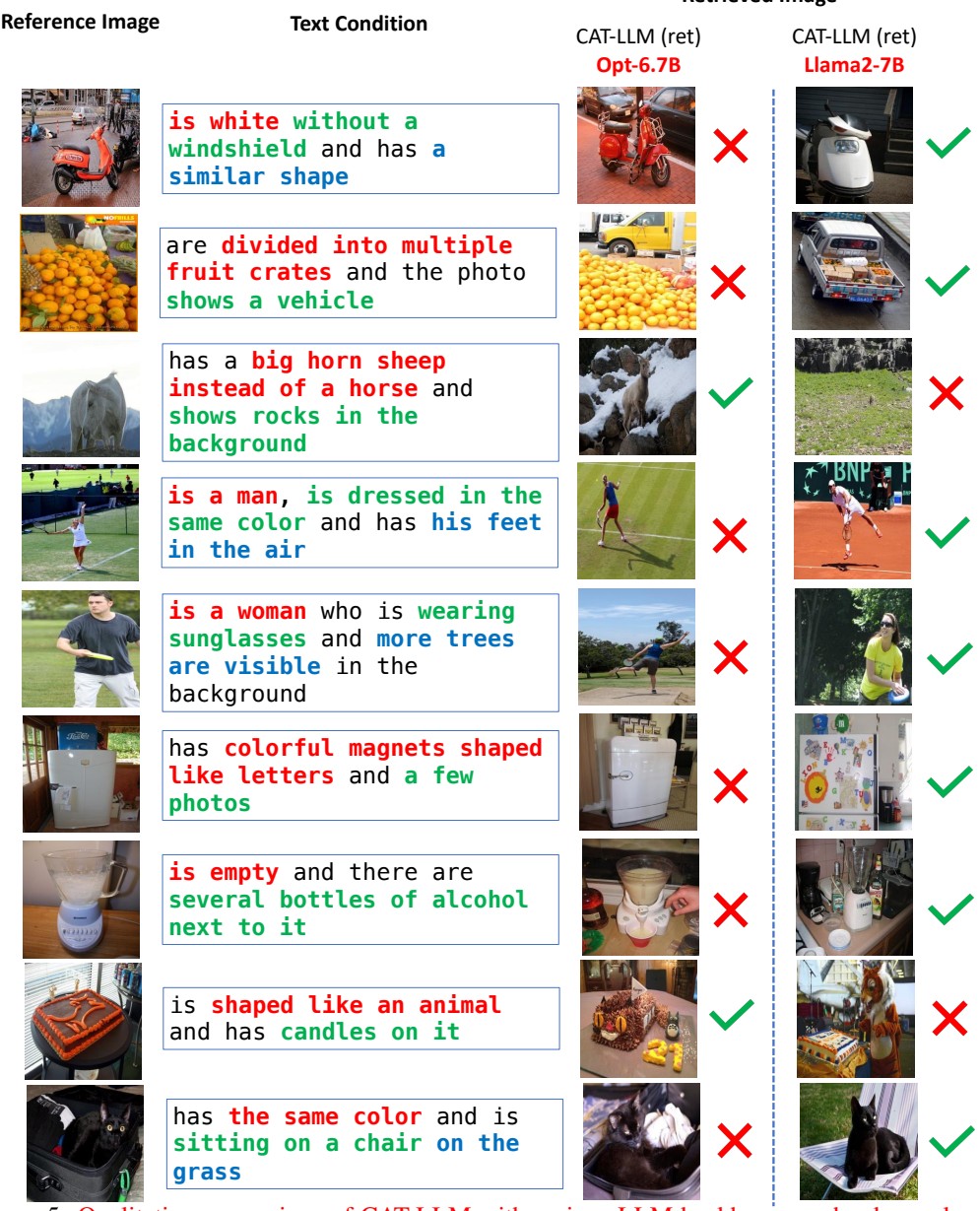

Figure 5: Qualitative comparison of CAT-LLM with various LLM backbones on hard samples selected from CIRCO test set. Critical textual cues in each sample are highlighted.

### A.3 FAILURE CASES STUDY

In Figure 6, we list several primary scenarios where CAT-LLM fails. The most significant is the quantities-related scenarios. When the text condition includes specific numerical requirements, such as 'two cats', CAT-LLM often struggles to retrieve target images with the correct number. Similarly, when the text condition contains words related to quantity, like 'fewer' or 'more', CAT-LLM also fails to accurately identify target images that correctly represent these quantitative relationships. This limitation mainly stems from CLIP's restricted capability in object counting, as detailed in (Radford et al., 2021). CAT-LLM utilizes CLIP's visual embeddings as its visual input and relies on CLIP's aligned image-text space for retrieval, resulting in CAT-LLM being restricted to the same limitations as CLIP in object counting. Additionally, CAT-LLM struggles with the text condition such as 'greyscale' and 'shot angle'. These text conditions, instead of relating to the image content, are associated with the image's state or attributes. Enhancing CAT-LLM to better address these types of requirements is one of our future work.

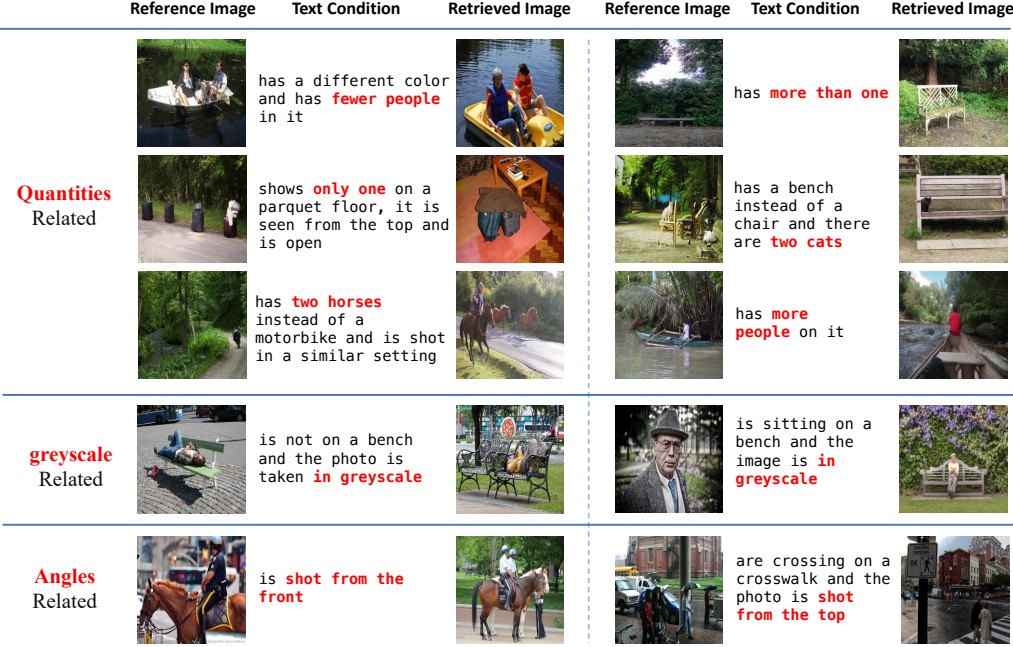

Figure 6: Failure cases of CAT-LLM.

### A.4 QUALITATIVE RESULTS ON CIRCO

Figure 7 shows more qualitative results from the CIRCO validation set. The evaluation samples from CIRCO are diverse and of high quality. Importantly, they provide multiple ground truth labels for each input, which helps in a more comprehensive analysis. From the figure, we observe that: (1) CAT-LLM can effectively handle the multi-modal input and retrieve the target image. (2) Some failed examples indicate that our method struggles to distinguish quantity. This is an area we aim to improve in the future. (3) Some false negative samples are highlighted in red. This is primarily due to during the label annotation, the authors leverage their proposed SEARLE method to coarsely filter out images from a large gallery, leading to missing some true positives, which can be well retrieved by CAT-LLM. These false negatives indicate that CAT-LLM has different preferences compared to conventional CLIP inversion-based methods, suggesting CAT-LLM's potential to refine existing benchmarks.

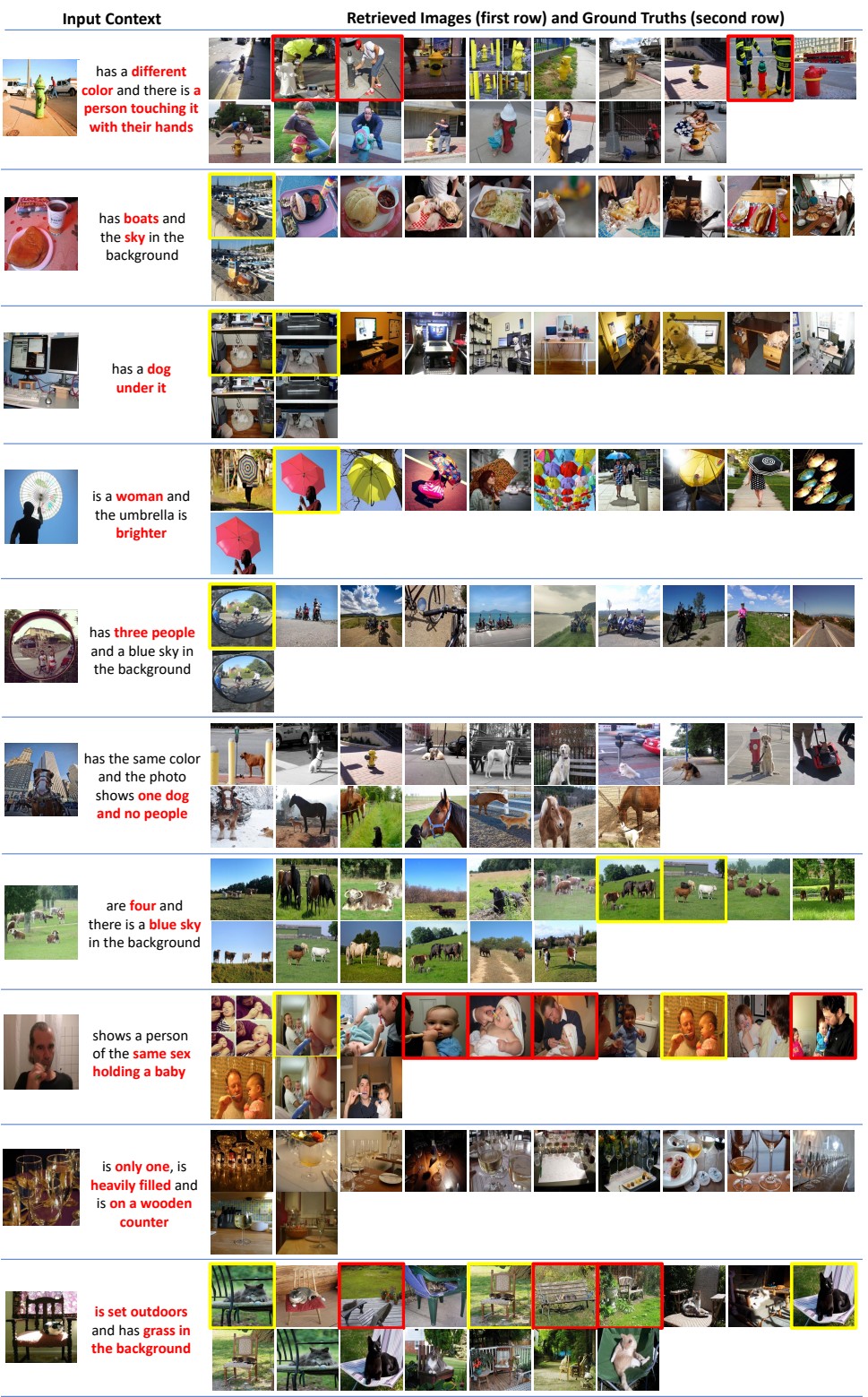

Figure 7: Qualitative results on CIRCO validation set. The first row is the ranked images retrieved by CAT-LLM-(ret), and the second row is the ground truth images. True positives are marked in yellow and false negatives are marked in red.

## B    LIMITATIONS

### B.1    LIMITATIONS INHERITED FROM CLIP MODEL

CAT-LLM leverage frozen CLIP model as the base image-text retrieval model. While it benefits from the strong aligned image-text space provided by CLIP, it also inherits CLIP's inherent limitations. For example, as shown in Figure 6, CAT-LLM struggles to retrieve the correct images in scenarios involving queries related to quantities, grey scale and angles. This issue stems from CLIP's intrinsic weaknesses, such as object counting, as detailed in Radford et al. (2021). It should be noted that another significant limitation of CLIP is the weak logical understanding ability of CLIP text encoder (Wang et al., 2023). Fortunately, as shown in Figure 4 CAT-LLM process the logical relationship in LLM space, thereby mitigating this issue.

### B.2    SLOW INFERENCE SPEED CAUSED BY LLM

A notable limitation of CAT-LLM is its inference speed, constrained by the language model. As shown in Table 9, we assessed the time consumed by different methods to conduct inference on 100 samples, with a batch size of 1. Due to the inherently slower inference speed of language models, CAT-LLM(ret) is slower than the conventional CIR model Combiner. The situation in CAT-LLM(cap) is worse because of the more time-consuming auto-regressive generation process. This time consumption attributed to the language model is one of the primary limitations of CAT-LLM. It should be noted that, different from the CIR methods like Combiner which can only process the query containing single image-text pair (i.e., CIR task), CAT-LLM can process dense scenarios containing multi-turns queries as demonstrated in Sec 4.2. This is one of the significant motivations that introduces an LLM into the MMCIR tasks.

Table 9: The time cost of processing 100 samples. The experiment is conduct on the same Nvidia A100 GPU.

| Method | CLIP image encoder | CLIP text encoder | LLM | combiner | All | FPS |
|---|---|---|---|---|---|---|
| Combiner | 1.1s | 1.1s | - | 0.1s | 2.3s | 43.5 |
| CAT-LLM-(ret) | 1.2s | - | 2.3s | - | 3.5s | 28.5 |
| CAT-LLM-(cap) | 1.2s | 1.0s | 25.0s | - | 27.2s | 3.7 |

### B.3    RETRIEVAL MECHANISM

In Section 4.1, we discovered that CAT-LLM-(ret) and CAT-LLM-(cap) has different biases towards the input context. CAT-LLM-(cap) is good at process the textual cues whereas CAT-LLM-(ret) excel at integrating image and text but tends to overlook some textual details. CAT-LLM-(ret+cap) combines the output of CAT-LLM-(ret) and CAT-LLM-(cap) achieves better results on CIRCO and CIRR benchmark. However, its performance on the more challenging benchmark GeneCIS shows a decline, indicating its limitations in specific scenarios. Additionally, the auto-regressive generation process in CAT-LLM-(cap) is time-consuming, posing challenges for real world applications. In the future, we aim to develop a more robust retrieval mechanism to efficiently extracting information from LLM space.

## C    DATA GENERATION

We employ the Llama2/7B-Chat model for our data generation. During each sample generation, we input the task description and an in-context example as the task prompt, as shown in Figure 8. It costs approximately 20 A100 GPU days to generate 1 million tuples, using image-caption pairs from CC3M (Sharma et al., 2018) as source pairs. We visualize some samples from MMC as shown in Figure 9. As we can see, Llama2 can generate diverse text conditions based on the source caption. The target caption effectively combines the source caption and the generated text condition.

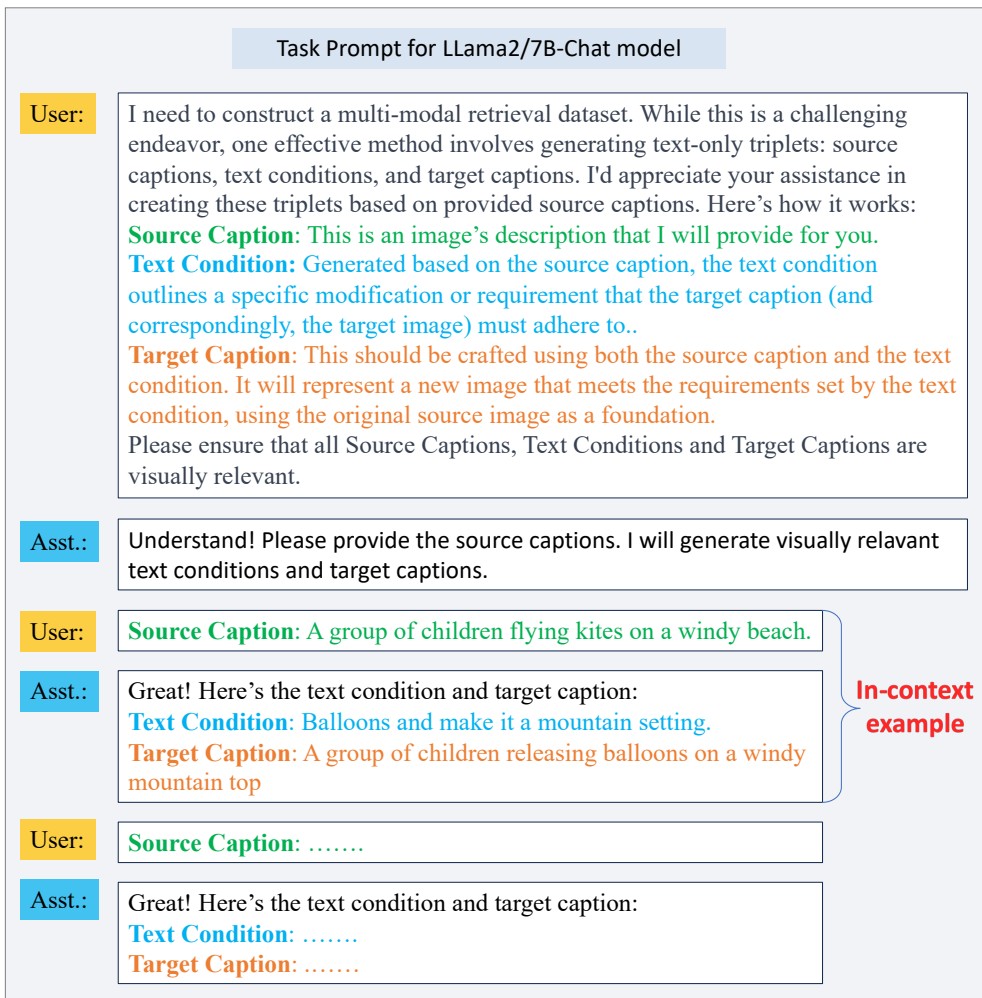

Figure 8: Our specialized task prompt for Llama2/7B-Chat model.

Table 10: Prompts for different tasks. The red text means the original input of this task. The blue text is the output of model.

| | Task / Objective | Text Condition Prompt | Retrieval / Captioning Prompt |
|---|---|---|---|
| Training | Cap | - | It is a photo of {caption} |
| | ITM | - | A photo of [ret] |
| | CA-Cap | Q: What if {text condition}? | A: It becomes a photo of {target caption} |
| | CA-ITM | Q: What if {text condition}? | A: It becomes a photo of [ret] |
| Inference | CIRCO | Q: What if {relative caption}? | A: It becomes a photo of [ret] |
| | CIRR | Q: What if {relative caption}? | A: It becomes a photo of [ret] |
| | GeneCIS (focus object) | Q: What if a similar scene with same {object}? | A: It becomes a photo of [ret] |
| | GeneCIS (change object) | Q: What if this scene appears {object}? | A: It becomes a photo of [ret] |
| | GeneCIS (focus attribute) | Q: What if this object with same {attribute}? | A: It becomes a photo of [ret] |
| | GeneCIS (change attribute) | Q: What if this object {attribute}? | A: It becomes a photo of [ret] |

# D   PROMPTS DESIGNING

In this paper, we instruct an LLM to multi-modal contextual image retrieval. We design various prompts for CAT-LLM to adapt to different tasks, as shown in Table 10.

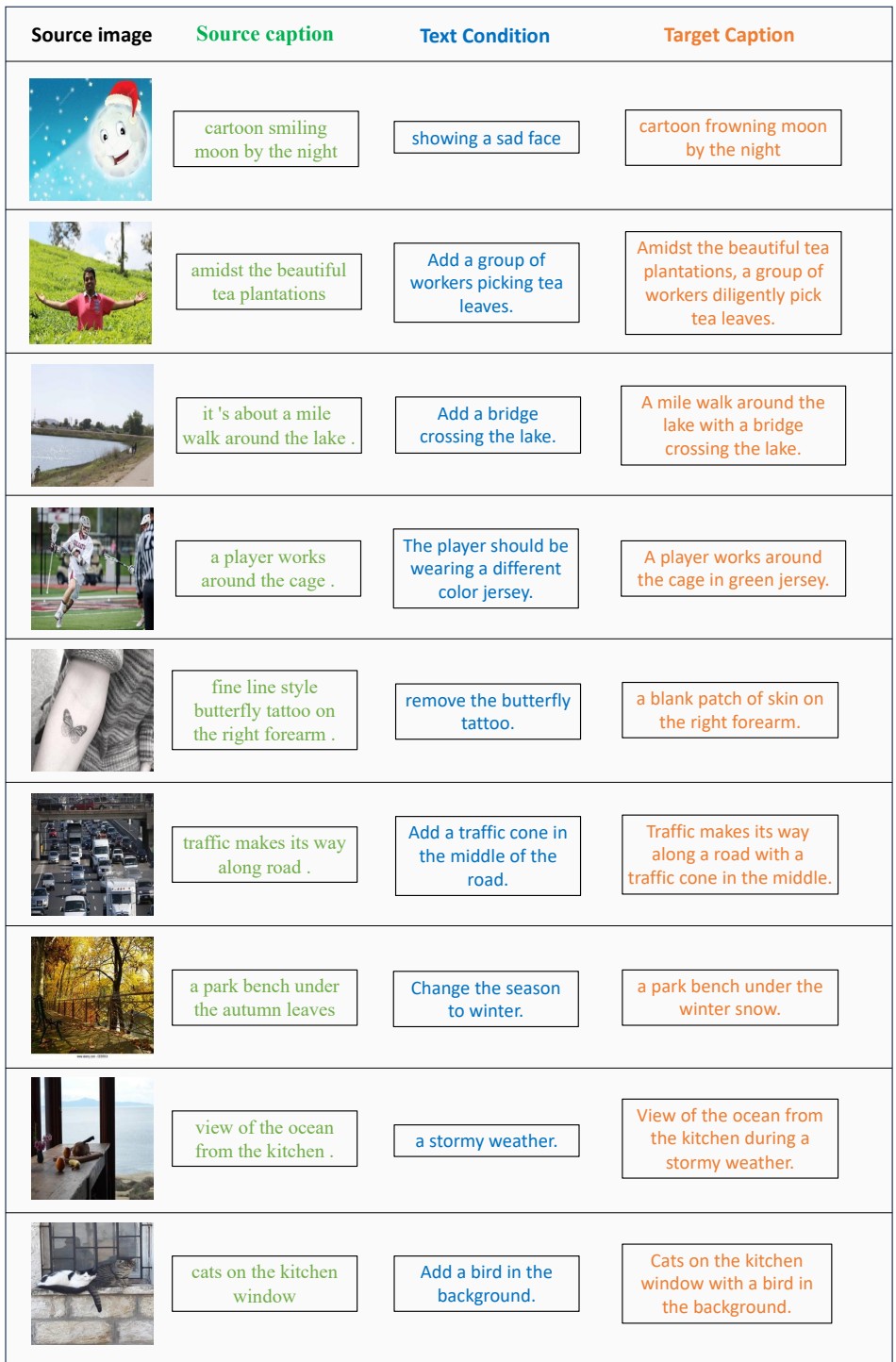

Figure 9: Data samples selected from MMC. Note that the source image is not visible to the language model during the text condition and target caption generation.

# E  INFERENCE ILLUSTRATION

The inference details are illustrated in 10.

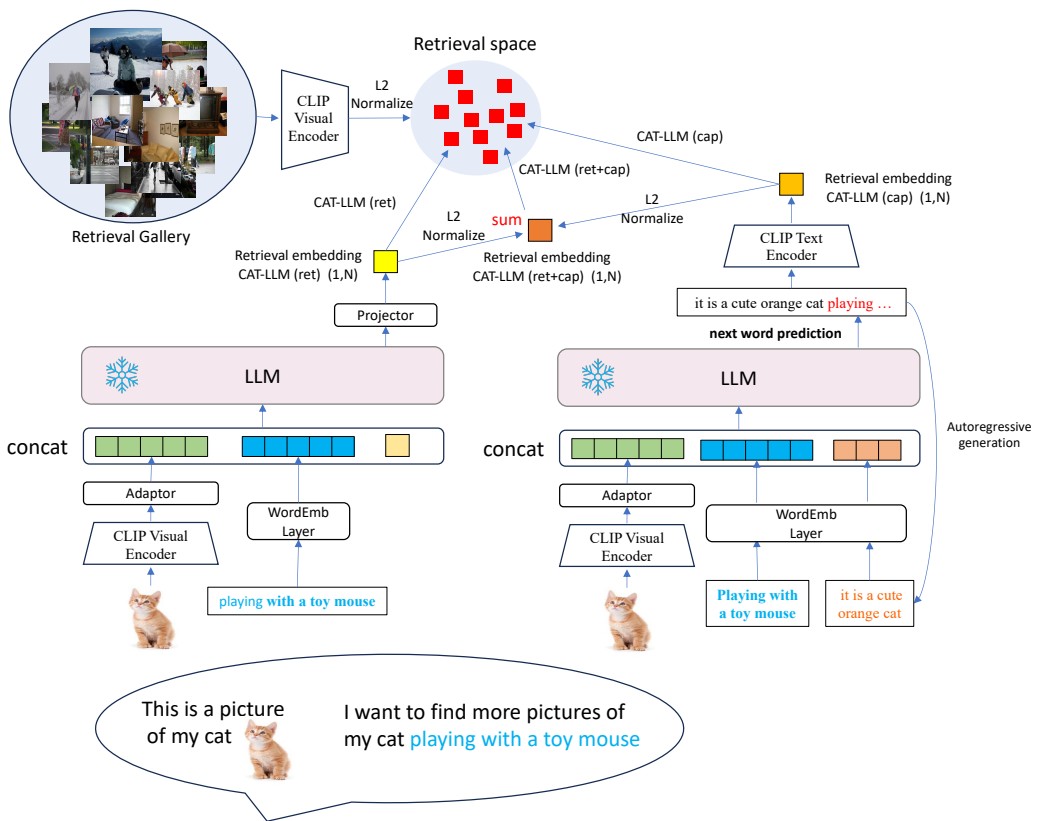

Figure 10: Inference illustration of CAT-LLM(ret), CAT-LLM(cap) and CAT-LLM(ret+cap).

Table 11: Quantitative results on CIRCO test set. We remind that the mAP@k metrics are computed considering all the ground truth images for each query, the Recall@k metrics are computed considering only the target image for each query (the one the author used to write the relative caption).

| Backbone | Method | mAP@K | | | | Recall@K | | | |
| --- | --- | --- | --- | --- | --- | --- | --- | --- | --- |
| | | $K=5$ | $K=10$ | $K=25$ | $K=50$ | $K=5$ | $K=10$ | $K=25$ | $K=50$ |
| L/14 | Image-only | 2.79 | 3.18 | 3.75 | 4.12 | 5.38 | 10.00 | 18.00 | 26.75 |
| | Text-only | 2.50 | 2.64 | 3.11 | 3.38 | 5.0 | 7.12 | 12.12 | 17.62 |
| | Image + Text | 6.37 | 7.04 | 8.11 | 8.72 | 14.37 | 21.12 | 34.75 | 46.25 |
| | Pic2Word | 8.72 | 9.51 | 10.64 | 11.29 | - | - | - | - |
| | SEARLE | 11.68 | 12.73 | 14.33 | 15.12 | 21.88 | 32.00 | 44.75 | 54.87 |
| | FROMAGe | 4.0 | 4.44 | 5.26 | 5.73 | 8.25 | 13.63 | 23.50 | 32.50 |
| | **CAT-LLM-(cap)** | 6.43 | 6.84 | 7.77 | 8.30 | 10.62 | 14.88 | 22.50 | 29.62 |
| | **CAT-LLM-(ret)** | _13.55_ | _14.70_ | _16.35_ | _17.28_ | **24.38** | _33.75_ | **46.25** | **59.00** |
| | **CAT-LLM-(ret+cap)** | **15.00** | **15.73** | **17.51** | **18.45** | _23.62_ | **34.50** | _45.62_ | _57.63_ |
| B/16 | Image-only | 1.30 | 1.74 | 2.21 | 2.52 | 4.50 | 8.00 | 15.25 | 21.75 |
| | Text-only | 2.59 | 2.75 | 3.12 | 3.30 | 4.75 | 6.88 | 10.88 | 14.88 |
| | Image + Text | 2.60 | 3.19 | 4.12 | 4.63 | 8.62 | 13.63 | 24.50 | 35.00 |
| | **CAT-LLM-(cap)** | 5.35 | 5.59 | 6.32 | 6.68 | 8.50 | 12.12 | 17.75 | 24.00 |
| | **CAT-LLM-(ret)** | _12.79_ | _13.28_ | _14.89_ | _15.65_ | _24.00_ | _31.75_ | **44.50** | **54.12** |
| | **CAT-LLM-(ret+cap)** | **13.95** | **14.47** | **16.00** | **16.74** | **25.00** | **31.87** | _44.12_ | _52.38_ |

# F COMPLETE RESULTS ON CIRCO

The Complete results including the recall@K metrics are shown on Table 11.

## G TRAINING DETAILS OF COMBINER

In this paper, we train the Combiner (Baldrati et al., 2022a) using proposed MMC dataset. Given the training data (ref image, text condition, target caption), the Combiner takes reference image and text condition as input. The ref image and text condition are first encoded by CLIP image encoder and CLIP text encoder, respectively. These two clip features are then composed into a single vector through a combiner component (MLPs). A contrasive loss is then used to align the output single vector and the target feature, i.e., the CLIP text feature of the target caption. The Combiner is trained on MMC for 6 epochs with a batchsize 1024. The temperature in contrastive loss in set to 15. The CLIP model is frozen during training. Other training parameters are the same as Vaze et al. (2023).

