# OpenReview forum: "CAT-LLM: Context-Aware Training enhanced Large Language Models for multi-modal contextual image retrieval"
_ICLR.cc/2024/Conference — ICLR 2024 Conference Withdrawn Submission_

### Official Review · Reviewer_34Fi · 2023-10-29

**Soundness:** 3 good
**Presentation:** 3 good
**Contribution:** 3 good
**Rating:** 5
**Confidence:** 4

**Summary:**

This paper proposes a method utilizing LLMs to tackle the Multi-Modal Contextual Image Retrieval (MMCIR) problem. The authors construct a Multi-Modal Captioning (MMC) dataset and introduce Context-Aware Captioning (CA-Cap) and Context-Aware Text Matching (CA-TM) objectives to train a frozen LLM for MMCIR. The proposed method has shown promising results on various benchmarks.

**Strengths:**

1. Since LLMs are good at processing and integrating contextual information, utilizing LLMs rather than text encoders derived from image-text matching models is promising to address the Multi-Modal Contextual Image Retrieval (MMCIR) problem.
2. The authors construct a Multi-Modal Captioning (MMC) dataset by enriching existing image captioning datasets from ⟨image, caption⟩
to ⟨reference image,reference caption,text condition,target caption⟩, which can be helpful.

**Weaknesses:**

1. The description of the inference is too brief to understand. How are the fused CAT-LLM-(ret) and CAT-LLM-(cap) used to retrieve images?
2. The authors mention that CLIP text encoder struggles with understanding objectrelations, word order and logic. However, the authors use the features of the target caption from CLIP text encoder to align with the ret token, and further utilize the representations of the generated caption from from CLIP text encoder for inference, which does not make sense to me.

**Questions:**

Please see Weaknesses.

**Details Of Ethics Concerns:**

In my opinion, no ethics review are needed.

---

> ### Author Response · Authors · 2023-11-18
> **Response to Reviewer 34Fi**
>
> > Q1: The description of the inference is too brief to understand. How are the fused CAT-LLM-(ret) and CAT-LLM-(cap) used to retrieve images?
>
> Ans: Both of CAT-LLM-(ret) and CAT-LLM-(cat) finally generate a retrieval embedding with dimensions (1,N). For CAT-LLM-(ret+cap) mode, we first normlize the retrieval embedding of CAT-LLM-(ret) and CAT-LLM-(cap). Then, we obtain the final retrieval embedding by simply summing these normalized embeddings. For a better understanding, we add a figure in Appendix E to illustrate the inference of CAT-LLM.
>
> > Q2: The authors mention that CLIP text encoder struggles with understanding objectrelations, word order and logic. However, the authors use the features of the target caption from CLIP text encoder to align with the ret token, and further utilize the representations of the generated caption from CLIP text encoder for inference, which does not make sense to me.
>
> Ans: Thank you for the careful review. This is indeed an interesting question worth discussing.
>
> 1. **Logical words in CIR tasks.**
>
>     Different from conventional image-text tasks that mainly involve caption-style texts, logical words appear more frequently in CIR task. The text conditions in CIR tasks serve to describe the difference between the reference and target images. As a result, logical words such as "no" and "instead of" are commonly used in these text conditions to indicate whether specific objects are present in the target image.
>
> 2. **The limitation of CLIP text encoder**.
>
>     CLIP text encoder excels at processing the caption style textual input , e.g., 'a photo of a black cat with flowers'. However, it struggles in textual input containing some logical words. For instance, when given the sentence like "a black cat is next to fruits instead of flowers", CLIP text encoder often struggles to understanding the phrase "instead of" and may ouput a representation that relates to both 'fruits' and 'flowers'. It is one of the limiations in previous CIR methods that leverages CLIP text encoder to handle the text condition.
>
> 3. **Why CAT-LLM can address this limatation**.
>
>     In CAT-LLM, we leverage the aligned CLIP image-text space as the retrieval space. The CLIP text encoder is utilized to process the target caption in MMC and the generated caption from CAT-LLM-(cap). In these two scenarios, the text inputs of CLIP text encoder are more likely in a caption-style. The text conditions, which are more likely to contain logical words, are processed in the language model space, where the logical words like 'not' and 'instead of' can be easily understood.
>
> 4. **Qualitative results to support above discussion**.
>
>    In Appendix A.1, we provide qualitative results to support the above discussion. We compare the results of Combiner and CAT-LLM towards the queries containing logical words. Combiner is a classic CLIP based CIR method. It leverages a simple combiner component to integrate the image and text features extracted from frozen CLIP model. Both of CAT-LLM and Combiner are trained on the same MMC dataset. For comparsion, we select samples from CIRCO validation set that include logical words like 'no', 'not' and 'instead of'.  As shown in Figure 4, CAT-LLM shows strong capability in understanding the logical words and retrieve the correct images. While the Combiner struggles to retrieve correct images due to the limitation of CLIP text encoder in understanding logical words. During our checking, there is only one exception where Combiner successfully retrieve the image while CAT-LLM  dose not (the last sample shown in the figure).
>
> 5. **Other limitations in CAT-LLM caused by CLIP encoder.**
>
>    Leveraging frozen CLIP as the base retrieval model indeed caused some limitations. As shown in Appendix A.3, we observe that CAT-LLM often struggles in handling the cases involving numbers or numerical words like "more" and "fewer". It is caused by the inherent limitations of the CLIP model in counting objects, as described in the CLIP paper.

---

### Official Review · Reviewer_xXn6 · 2023-11-01

**Soundness:** 3 good
**Presentation:** 2 fair
**Contribution:** 2 fair
**Rating:** 5
**Confidence:** 4

**Summary:**

This paper proposes a method that employs LLMs to address the Multi-Modal Contextual Image Retrieval (MMCIR) problem. Specifically, authors construct a Multi-Modal Captioning (MMC) dataset with CC3M and Llama2, and introduce two another objectives, including a Context-Aware Captioning (CA-Cap) and a Context-Aware Text Matching (CA-TM) objective. The CA-Cap aims to predict the next target token conditioned on the mapped visual vectors, text condition tokens and previous target tokens. The CA-TM is an info-NCE loss maximizing the similarity between the ret token and clip features of target caption. The trained frozen LLM achieve competitive results on several image-language tasks like Zero-Shot Composed Image Retrieval (ZS-CIR), Visual Storytelling and Visual Dialog.

**Strengths:**

1) Construct an MMC dataset based on the off the shelf CC3M dataset, with Llama2 and in context learning.
2) Utilize the decoder-only LLM for Retrieval tasks with frozen CLIP image and text encoders, by introducing two tasks: Loss_cap for capion generation with LLM; Loss_itm for image-text matching with a retrieval token appended at the end of the input tokens.
3) Design two new objectives (CA-TM and CA-Cap) for MMCIR tasks. The experiments show that these two objectives improve performances in zero-shot composed image retrieval and dense multi-modal contextual retrieval.

**Weaknesses:**

1) The quality of generated <T_con, T_tgtc> cannot be guaranteed, and a process for filtering and checking (manually or automatically) is necessary.
2) Compared with standard Loss_cap and Loss_itm, the “context-aware” in CA-Cap and CA-TM only seems like an augmentation of data that enriches and extends the details of input texts. Do the improvements come from the more detailed text inputs from the new dataset or the two new objectives? Will baseline and competing methods perform better when trained with the newly proposed dataset in this paper?
3) On the evaluation of CIRCO and GeneCIS, the metrics Recall@K and Avg R@1 have a performance decline when CAT-LLM-(ret) is added with CAT-LLM-(cap). This phenomenon is lack of analysis.
4) Since this paper proposes to perform the MMCIR task using LLMs, it’s necessary to compare the results of various LLMs with different sizes (e,g., OPT-2.7B, Llama-7B, (FLAN) T5-(X)XL).
5) This paper only encodes the image into a set of prefix prompt tokens, just like what recently proposed Multimodal Large Language Models (MLLM, e.g., BLIP-2, LLaVA, mPLUG-Owl) do, and these MLLMs are also compatible with the methods in this paper. I think they may perform better when fine-tuning with Loss_cap and Loss_itm here since their poweful ability of image-langauge understanding.

**Questions:**

All questions are included in the weakness section.

---

> ### Author Response · Authors · 2023-11-18
> **Response to Reviewer xXn6 (1/2)**
>
> > Q1: The quality of generated <T_con, T_tgtc> cannot be guaranted, and a process for filtering and checking (manually or automatically) is necessary.
>
> Ans: We would like to thank the reviewer for the suggestion for data filtering. Designing a filtering strategy for the MMC dataset is both interesting and valuable, particularly because: (1) it is genertaed from LLM, different from previous noise data collected from web. (2) The data structure in MMC is unique, containing <I_ref, T_refc, T_con, T_tgtc>. The filtering strategy on this structured data can be very different from previous image-caption filtering strategy. We leave it as our future work.
>
> In response to the current MMC dataset, we offer the following responses:
>
> 1. **Our efforts to improve the quality of generated data.** We have manually checked the quality of the generated data at the early stage and adjust the task prompt used in data generation. We found that the in-context examples can largely improve the quality of generated data.
>
> 2. **Data Scale.**  We have not designed more filtering strategies on the <I_ref, T_con, T_tgtc> triplets except for excluding samples contains very long texts. One of the reason is that the data scale of MMC is 1M, that helps mitigate the impact of bad samples.
>
> 3. **The quality of MMC dataset.** To investigate the quality of MMC dataset, we train a Combiner model using the MMC dataset. Combiner is a classic CLIP based CIR method. It leverages a combiner component to integrate the image and text features extracted from frozen CLIP model.
>
>    As the table shown, the Combiner trained on our generated MMC dataset achieves comparable results, approaching previous sota method SEARLE. It demonstrates the effectiveness of our MMC dataset.
>
>    | Methods       | Training data | mAP@5 | mAP@10 | mAP@25 | mAP@50 |
>    | ------------- | ------------- | ----- | ------ | ------ | ------ |
>    | Pic2Word      | CC3M          | 8.72  | 9.51   | 10.64  | 11.29  |
>    | SEARLE        | ImageNet1K    | 11.68 | 12.73  | 14.33  | 15.12  |
>    | Combiner      | MMC           | 11.63 | 12.46  | 13.79  | 14.60  |
>    | CAT-LLM-(ret) | MMC           | 13.55 | 14.70  | 16.35  | 17.28  |
>
> > Q2: Compared with standard Loss_cap and Loss_itm, the “context-aware” in CA-Cap and CA-TM only seems like an augmentation of data that enriches and extends the details of input texts. Do the improvements come from the more detailed text inputs from the new dataset or the two new objectives? Will baseline and competing methods perform better when trained with the newly proposed dataset in this paper?
>
> Ans:
>
> 1. **MMC dataset.** We construct triplets <ref image, text condition, target caption> from <ref image, ref caption>. The triplets in MMC are different from the text augmentation that enriching the caption to long and detailed discription.
> 2. **Context-aware objectives.** Our proposed objectives CA-Cap and CA-TM are intergrated with the MMC dataset. Conventional Cap and ITM loss only "translate" the input from one modality to another modality in LLM space (i.e., from image to caption and from caption to image). Our proposed context-aware objectives require the LLM extract target information (i.e., target caption) by considering the context (i.e., the ref image and the text condition). These context-aware objectives enable the LLM to more effectively integrate the multi-modal input, leading to improved performance on MMCIR tasks, that invovle multi-modal inputs.
> 3. **Visualization of the effectiveness of context-aware training.** To better illustrate the benefits of our proposed context-aware training, we have added a visualization of token relevance in LLM space in **Figure 3**. The figure shows that our context-aware training enables the model effectively integrating the visual input and key textual cues to accurately retrieve the target image.  The model without context-aware training tends to concentrate on visual tokens or with only partial textual cues, leading to incorrect retrieval results.
>
> 4. **CAT-LLM benefits from (1) Context-aware training (the proposed objectives and the proposed dataset):** As shown in Table 6 and Figure 3, the context-ware training largely improves the performance of CAT-LLM. **(2) The language model**. To investigate the impact of MMC on conventional methods, we train a classic CIR method, i.e., Combiner using MMC dataset. Table 8 shows the results. The Combiner model trained on the MMC dataset achieves comparable results, approaching previous textual inversion-based method SEARlE. **Although trained on the same MMC dataset, there is still a significant gap between the Combiner and CAT-LLM, demonstrating the effectiveness of LLM in MMCIR tasks**. In Appendix A.1, we provide additional qualitative results and analyses on the differences between CLIP based Combiner and CAT-LLM in processing logical words.
>
> 5. **MMC dataset for other methods.** We report the results of Combiner trained on MMC. Table 8 shows the results.

---

> ### Author Response · Authors · 2023-11-18
> **Response to Reviewer xXn6 (2/2)**
>
> > Q3: On the evaluation of CIRCO and GeneCIS, the metrics Recall@K and Avg R@1 have a performance decline when CAT-LLM-(ret) is added with CAT-LLM-(cap). This phenomenon is lack of analysis.
>
> Ans:
>
> 1. **CIRCO Recall@K.** We'd like to remind that there multiple ground truth for each query in CIRCO benchmark. The mAP metric that considers all the ground truth is a robust metric against ReCall@K. CAT-LLM achieves consistent results on mAP@K in Table 1 and Table 7 (newly add) demonstrates it robust on CIRCO benchmark. We should remind that the Recall@K metric in CIRCO are computed considering only the target image for each query (the one the author used to write the relative caption), rather than computed the first true positve among the multiple ground truth. To avoid the potential misunderstanding of Recall@K metric in CIRCO, we move the Recall@K results to Appendix for future reference.
>
> 2. **GeneCIS R@1.** CAT-LLM-(ret+cap) achieves lower performance than CAT-LLM-(ret) on GeneCIS which is inconsistent with the results observed in CIRCO and CIRR. We think this phenomenonis primarily caused by **the less informative textual input in GeneCIS.** In CIRCO and CIRR, the text condition is a relative caption containing many information about the target image. In GeneCIS, the text input is merely an object name or an attribute, offering less information. As disscussed in Sec 4.1, the CAT-LLM-(cap) method is inclined to capture the textual information. When the textual information is limited, it is hard the enhance the CAT-LLM-(ret) by simply summing them.
>
> > Q4: Since this paper proposes to perform the MMCIR task using LLMs, it’s necessary to compare the results of various LLMs with different sizes (e,g., OPT-2.7B, Llama-7B, (FLAN) T5-(X)XL).
>
> Ans: Thank you for the suggestion. We added a ablation about the LLM backbones on Table 7, Sec 4.3. As shown in the table, CAT-LLM can effectively intergrated with various LLMs, gaining from model size and stronger model. The powerful language model Llama2-7B, markedly improve the performance of CAT-LLM. To illustrate the impact of a advanced LLM backbone, we have added a qualitative comparison between CAT-LLM-Opt and CAT-LLM-Llama2 on challenging test samples in Appendix A.2.
>
> | LM backbone | Mode    | mAP@5 | mAP@10 | mAP@25 | mAP@50 |
> | ----------- | ------- | ----- | ------ | ------ | ------ |
> | Opt-6.7B    | cap     | 6.43  | 6.84   | 7.77   | 8.30   |
> |             | ret     | 13.55 | 14.70  | 16.35  | 17.28  |
> |             | ret+cap | 15.00 | 15.73  | 17.51  | 18.45  |
> | Opt-2.7B    | cap     | 6.64  | 7.16   | 7.95   | 8.44   |
> |             | ret     | 12.27 | 13.09  | 14.66  | 15.50  |
> |             | ret+cap | 14.05 | 14.87  | 16.52  | 17.35  |
> | Llama2-7B   | cap     | 8.87  | 9.63   | 10.90  | 11.52  |
> |             | ret     | 16.00 | 16.82  | 18.61  | 19.59  |
> |             | ret+cap | 17.28 | 18.27  | 20.22  | 21.17  |
>
>
> > Q5: This paper only encodes the image into a set of prefix prompt tokens, just like what recently proposed Multimodal Large Language Models (MLLM, e.g., BLIP-2, LLaVA, mPLUG-Owl) do, and these MLLMs are also compatible with the methods in this paper. I think they may perform better when fine-tuning with Loss_cap and Loss_itm here since their poweful ability of image-langauge understanding.
>
> Ans:
>
> We agree that finetuning recently proposed MLLMs with our proposed dataset and objectives is likely to largely improve the performance on MMCIR benchmarks, due to the better image-text backbone (BLIP2) and the enhanced image-language understanding.
>
> However, leveraging these MLLMs (e.g., BLIP2, LLaVA, mPLUG-Owl) may conflict with the zero-shot setting. The zero-shot CIR benchmarks like CIRCO and GeneCIS leverage the images sourced from MSCOCO and Visual Genome to construct their test samples. While most recently MLLMs uses data from MSCOCO for pretraining. E.g., BLIP2 uses the dataset including MSCOCO and Vistal Genome for pretraining. LLaVA uses image-caption data from MSCOCO as the source data to generate an instruction dataset. mPLUG-Owl leverages MSCOCO for pretraining and using the data from LLaVA for instruciton tuning. This overlap raises concerns about the potential impact of using images from the same domain as the training data, which could largely affect the model's performance in a zero-shot setting.
>
> The competing methods in this paper like Pic2Word, Combiner-(GeneCIS) and Formage are trained on CC3M dataset or on generated data sourced from CC3M. Based on this, we generate MMC using CC3M as the data source for better comparion and analysis.
>
> It is valuable to explore the impact of employing a more powerful MLLM on solving MMCIR tasks. We consider it as our future work.

---

> > ### Comment · Reviewer_xXn6 · 2023-12-03
> >
> > Thanks for the authors' response, after carefully checking the rebuttal and the arguments of the authors, doubts remain regarding the statements and claims in the manuscript, particularly in relation to the “context-aware”. I intend to keep my rating.

---

### Official Review · Reviewer_USW9 · 2023-11-01

**Soundness:** 2 fair
**Presentation:** 3 good
**Contribution:** 2 fair
**Rating:** 5
**Confidence:** 4

**Summary:**

The authors propose a solution for the task of Multi-modal composite image retrieval, leverage the potential of Large Language models. imoprtantly they cater to the requirement of multiple image and text queries at input for retrieval. Specifically, they introduce two objectives of Context-aware captioning and Contxt-aware text matching for context-aware training of an LLM for retrieval. Furthermore they also introduce a multi-modal captioning dataset to enhance training.

**Strengths:**

- The concept introduced here is really interesting, although the components used here carry less novelty.

- The writing of introduction, and the overall paper is quite fluid and easy to understand.

- The idea of using captioning as a task for context-awareness is well formulated.

- Qualitative Figures are well portrayed.

**Weaknesses:**

- Given that sufficient experiments are conducted, little reasoning is provided as to why the methods perform (low/high) in the way they do. More analytical reasoning would be encouraged.

- Does this retrieval include images containing multiple target objects for retrieval as well?

- More ablations on design choices, and not just learning objectives would have been more insightful.

- A time complexity analysis would have been helpful to understand the real-world adoption of such a method

- How significant do the authors expect the newly introduced dataset to be for the community as it could be easily generated as shown in the paper? Maybe other researches may modify on this synthesising process to get the data they need instead of using the proposed dataset ?

**Questions:**

- What about the time complexity of the proposed method against state-of-the-art methods ?
- What are some of the limitations of this method ?

---

> ### Author Response · Authors · 2023-11-18
> **Response to Reviewer USW9 (1/2)**
>
> > Q1: Given that sufficient experiments are conducted, little reasoning is provided as to why the methods perform (low/high) in the way they do. More analytical reasoning would be encouraged.
>
> Ans: Thank you for your suggestion. We have included following content in our manuscript to analyze the reasons behind the performance of CAT-LLM.
>
> 1. We add a comparion between CAT-LLM and the Combiner in Table 8, Sec 4.3.  Combiner is a classic CLIP based CIR method. It leverages a combiner component to integrate the image and text features extracted from frozen CLIP model. Both of CAT-LLM and Combiner are trained on the same MMC dataset.
>
>    | Methods       | Training data | mAP@5 | mAP@10 | mAP@25 | mAP@50 |
>    | ------------- | ------------- | ----- | ------ | ------ | ------ |
>    | Pic2Word      | CC3M          | 8.72  | 9.51   | 10.64  | 11.29  |
>    | SEARLE        | ImageNet1K    | 11.68 | 12.73  | 14.33  | 15.12  |
>    | Combiner      | MMC           | 11.63 | 12.46  | 13.79  | 14.60  |
>    | CAT-LLM-(ret) | MMC           | 13.55 | 14.70  | 16.35  | 17.28  |
>
>    (1) **The effectiveness of proposed MMC dataset.** As the table shown, the Combiner trained on our generated MMC dataset achieves comparable results, approaching previous sota method SEARLE. It demonstrates the effectiveness of our MMC dataset.
>
>    (2) **The effectiveness of LLM in MMCIR tasks.** Compare the results of Combiner and CAT-LLM, we can find that there is still a performance gap although trained on the same dataset. It demonstrates the effectiveness of LLM in MMCIR tasks. We provide more qualitative results on Appendix A.1 to illustrate the significant difference of CAT-LLM and CLIP based Combiner towards the text condition containing logical words.
>
>    | Methods       | Training data | mAP@5 | mAP@10 | mAP@25 | mAP@50 |
>    | ------------- | ------------- | ----- | ------ | ------ | ------ |
>    | Pic2Word      | CC3M          | 8.72  | 9.51   | 10.64  | 11.29  |
>    | SEARLE        | ImageNet1K    | 11.68 | 12.73  | 14.33  | 15.12  |
>    | Combiner      | MMC           | 11.63 | 12.46  | 13.79  | 14.60  |
>    | CAT-LLM-(ret) | MMC           | 13.55 | 14.70  | 16.35  | 17.28  |
>
> 3. **The effectiveness of our context-aware training.** We added a visualization to illustrate the effectiveness of our context-aware training in Figure 3, Sec 4.3. As the figure shown, our context-aware training enables the LLM attention to the both visual and language inputs, and well conbined the multi-modal context to retrieve the correct image.
>
> 4. **The failure scenarios of CAT-LLM**
>
>    We added a faliure cases study on Appendix A.3. As the figure shown, CAT-LLM struggles in quantities related cases. It can not identifies the number like 'two cats' and also the 'fewer' and 'more'. This limitation mainly stems from CLIP's weak capability in object counting, as described in CLIP paper. Additionally, we find that CAT-LLM struggles in cases retlated to 'greyscale' and  'angles'. These types of cases is not related the image content but the state or attributes of the image itself.
>
> > Q2: Does this retrieval include images containing multiple target objects for retrieval as well?
>
> Ans: On CIRCO benchmark, there is multiple ground truthes for each query, and we report a fine-grained metric mAP@K that consider all the ground truth on this benchmark. We provide more quaiitative results and analysis of CIRCO's multiple ground truth on  Appendix A.4. As for other benchmarks, such as CIRR and GeneCIS, there is only one ground truth for one query.
>
> > Q3: More ablations on design choices, and not just learning objectives would have been more insightful.
>
> Ans: Thanks for you suggestion. We added a ablation about the LLM backbones on Table 7, Sec 4.3.
>
> | LM backbone | Mode    | mAP@5 | mAP@10 | mAP@25 | mAP@50 |
> | ----------- | ------- | ----- | ------ | ------ | ------ |
> | Opt-6.7B    | cap     | 6.43  | 6.84   | 7.77   | 8.30   |
> |             | ret     | 13.55 | 14.70  | 16.35  | 17.28  |
> |             | ret+cap | 15.00 | 15.73  | 17.51  | 18.45  |
> | Opt-2.7B    | cap     | 6.64  | 7.16   | 7.95   | 8.44   |
> |             | ret     | 12.27 | 13.09  | 14.66  | 15.50  |
> |             | ret+cap | 14.05 | 14.87  | 16.52  | 17.35  |
> | Llama2-7B   | cap     | 8.87  | 9.63   | 10.90  | 11.52  |
> |             | ret     | 16.00 | 16.82  | 18.61  | 19.59  |
> |             | ret+cap | 17.28 | 18.27  | 20.22  | 21.17  |
>
> As shown in the table, CAT-LLM can effectively intergrated with various LLMs, gaining from model size and stronger model. The powerful language model Llama2-7B, markedly improve the performance of CAT-LLM. We provide more qualitative results on Appendix A.2 to illustarte the impact of a stronger language model backbone.

---

> ### Author Response · Authors · 2023-11-18
> **Response to Reviewer USW9 (2/2)**
>
> > Q4: A time complexity analysis would have been helpful to understand the real-world adoption of such a method
>
> Ans: Thank you for your suggestion, we have added a time complexity analysis on Appendix B.2. The experiment is conducted on the same A100 device. We measure the time cost of different methods processing 100 queries.
>
> | Methods       | CLIP image encoder | CLIP text encoder | LLM   | combiner | All   | FPS  |
> | ------------- | ------------------ | ----------------- | ----- | -------- | ----- | ---- |
> | Combiner      | 1.1s               | 1.1s              | -     | 0.1s     | 2.3s  | 43.5 |
> | CAT-LLM-(ret) | 1.2s               | -                 | 2.3s  | -        | 3.5s  | 28.5 |
> | CAT-LLM-(cap) | 1.2s               | 1.0s              | 25.0s | -        | 27.2s | 3.7  |
>
> As shown in the table, CAT-LLM is slower than the conventional CIR method Combiner, due to the inherently slower inference speed of LMs. CAT-LLM-(cap) is more time-consuming due to the auto-regressive generation process. We have included this time complexity into the limiations of CAT-LLM. It should be noted that, different from prior CIR methods like Combiner which can only process the query containing single image-text pair (i.e., CIR task), CAT-LLM can process dense scenarios containing multi-turns queries as demonstrated in Sec 4.2. This is one of the significant motivations that introduces an LLM into the MMCIR tasks.
>
> > Q5: How significant do the authors expect the newly introduced dataset to be for the community as it could be easily generated as shown in the paper? Maybe other researches may modify on this synthesising process to get the data they need instead of using the proposed dataset ?
>
> 1. **MMC dataset for MMCIR tasks**. In this paper, we have demonstrated that MMC dataset can be used for LLM context-aware training and conventional CIR model training. We will release this dataset for future research on this field.
>
> 2. **To modify the synthesising process**. It is feasible for other researchers to adapt and modify our synthesis process to create diverse, high-quality datasets tailored to their specific needs. Our dataset generation process, which utilizes the open-sourced Llama2 chat model and the CC3M dataset, is easy to follow.
>
> 3. **Leveraging LLM to generated structured vision-language data.**  The robust text generation capabilities of LLMs enable the generation of a wide range of specialized, structured text data to meet specific needs. For instance, In CAT-LLM, we utilize LLM to generate <target caption, reference caption>. By integrating it with the reference image, we obtain structed triplets <ref image, target caption, reference caption> for CAT-LLM context-aware training. We hope our dataset generation process, that intergrate images with structured text data generated by LLMs, could inspire the research in domains where specialized structed vision-language data is required yet hard to collect.
>
> > Q6: What about the time complexity of the proposed method against state-of-the-art methods ?
>
> Answered in Q4.
>
> > Q7: What are some of the limitations of this method ?
>
> Ans: We conclude the limitations of CAT-LLM on Appendix B. Mainly including:
>
> 1. **Limitations inherited from CLIP model**. CAT-LLM leverages CLIP feature space as the image-text retrieval space, resulting some limitations inheriting from CLIP, such as the object counting issue.
> 2. **Slow inference speed caused by LLM**. CAT-LLM is slower than conventional CIR mothods due to the lower inference speed of LLM.
> 3. **Univeral retrieval methanism**. In this paper, we propose CAT-LLM-(ret), CAT-LLM-(cap) and CAT-LLM-(ret+cap) for retrieval. In the future, we aim to develop a more robust univeral retrieval mechanism to efficiently extracting information from LLM space.

---

> ### Author Response · Authors · 2023-11-21
> **Response to Reviewer USW9 (Additions to Q2)**
>
> Q2: Does this retrieval include images containing multiple target objects for retrieval as well?
>
> Ans:
>
> In the previous response, we may have some misunderstanding about 'multiple target objects'. If by this term 'multiple target objects' you are referring to a query containing a single image and multiple text inputs, then this format is **not** present in current CIR benchmarks. In the CIR benchmarks, like CIRCO and CIRR, each query is a single image with a sentence, but the sentence may include multiple requirements. For instance, as illustrated in Figure 13 of Appendix A.2, there are challenging samples with complex text conditions, such as "**is white without a windshield** and **has a similar shape**".
>
> **CAT-LLM supports various multi-modal queries.** The LLM enables CAT-LLM to process more complex multi-modal scenarios, although without special training. In Sec 4.2, we demonstrate that CAT-LLM can handle queries containing multiple images and texts (VisualStory Telling) and multi-turns dialogs (Visual Dialog).

---

### Author Response · Authors · 2023-11-18
**General response**

We would like to thank the reviewers for their careful and constructive comments. The paper has been revised in accordance with the reviewers’ comments and suggestions. Updates and changes are marked by **red** color in the revised version. The major changes in this revision lie in following aspects:

1. **Context-aware training (Sec 4.3; Figure3).**

   To better illustrate the benefits of our proposed context-aware training, we have added a visualization of token relevance in LLM space that offers a more comprehensive understanding.

2. **CAT-LLM with various LLM backbone (Sec 4.3; Table 7)**.

   To study the impact of the LLM backbone on CAT-LLM, we have included an ablation study about CAT-LLM with various LLM backbone.

3. **MMC dataset on conventional methods (Sec 4.3; Table 8)**.

   To investigate the impact of MMC on conventional methods, we train a classic CIR method Combiner using MMC dataset.

4. **Analysis of the logical words in text condition (Appendix A.1; Figure 4)**.

   To illustrate CAT-LLM's enhanced ability in understanding text with complex logical relationships compared to the CLIP text encoder, we add the qualitative results of CAT-LLM and Combiner on scenarios involving logical words.

5. **Qualitative comparison of CAT-LLM with various LLM backbones (Appendix A.2; Figure 5).**

   To illustrate the impact of a advanced LLM backbone, we have added a qualitative comparison between CAT-LLM-Opt and CAT-LLM-Llama2 on challenging test samples.

6. **Analysis of failure cases in CAT-LLM (Appendix A.3; Figure 6)**.

   To explore and understand the limitations of CAT-LLM, we have added a failure case study.

7. **Limitations (Appendix B)**.

   We summarize the limations of CAT-LLM, mainly about (a) Limitations inherited from CLIP model. (2) Slow inference speed caused by LLM. . (3) Robust retrieval mechanism.

We hope these updates would address the concerns. Should you need further information, please let us know. We look forward to hearing from you soon.

---

> ### Author Response · Authors · 2023-11-22
> **A kind reminder regarding our response**
>
> We thank you for your time and effort in reviewing our paper. We have responded to all comments in our rebuttal. We would like to remind that the rebuttal period is approaching its end. If you have any other comments or questions, please let us know.
>
> Thank you for your attention. Sincerely Authors